# Notch controls the cell cycle to define leader versus follower identities during collective cell migration

Zain Alhashem[1], Dylan Feldner-Busztin[2], Christopher Revell[2], Macarena Alvarez-Garcillan Portillo[1], Karen Camargo-Sosa[3], Joanna Richardson[1], Manuel Rocha[4], Anton Gauert[1], Tatianna Corbeaux[1], Martina Milanetto[5], Francesco Argenton[5], Natascia Tiso[5], Robert N Kelsh[3], Victoria E Prince[4,6], Katie Bentley[2,7]*, Claudia Linker[1]*

[1]Randall Centre for Cell and Molecular Biophysics, Guy's Campus, King's College London, London, United Kingdom; [2]Cellular Adaptive Behaviour Lab, Francis Crick Institute, London, United Kingdom; [3]Department of Biology & Biochemistry, University of Bath, Bath, United Kingdom; [4]Committee on Development, Regeneration and Stem Cell Biology, The University of Chicago, Chicago, United States; [5]Department of Biology, University of Padova, Padova, Italy; [6]Department of Organismal Biology and Anatomy, The University of Chicago, Chicago, United States; [7]Department of Informatics, King's College London, London, United Kingdom

*For correspondence:
katie.bentley@crick.ac.uk (KB);
claudia.linker@kcl.ac.uk (CL)

**Abstract** Coordination of cell proliferation and migration is fundamental for life, and its dysregulation has catastrophic consequences, such as cancer. How cell cycle progression affects migration, and vice versa, remains largely unknown. We address these questions by combining in silico modelling and in vivo experimentation in the zebrafish trunk neural crest (TNC). TNC migrate collectively, forming chains with a leader cell directing the movement of trailing followers. We show that the acquisition of migratory identity is autonomously controlled by Notch signalling in TNC. High Notch activity defines leaders, while low Notch determines followers. Moreover, cell cycle progression is required for TNC migration and is regulated by Notch. Cells with low Notch activity stay longer in $G_1$ and become followers, while leaders with high Notch activity quickly undergo $G_1/S$ transition and remain in S-phase longer. In conclusion, TNC migratory identities are defined through the interaction of Notch signalling and cell cycle progression.

## Editor's evaluation

Using a combination of in vivo and in silico approaches, the authors have demonstrated how cell-fate decisions are orchestrated at the level of leader vs. follower cells in collective cell migration of trunk neural crest cells. They highlight the role of Notch signaling and cell cycle progression, showing how these traits differ between the leader and follower cells. The findings are of wide interest, as collective cell migration is a fundamental process critical for embryonic development as well as invasion of various cancers.

## Introduction

The harmonious coupling of cell proliferation with migration is fundamental for the normal growth and homeostasis of multicellular organisms. A prominent consequence of the dysregulation of these processes is cancer. Uncontrolled cell proliferation leads to primary tumours, and the acquisition of

migratory capacities leads to the formation of secondary tumours, the most common cause of cancer deaths. Metastatic cells can migrate collectively, which endows them with more aggressive behaviours (*Nagai et al., 2020*). Collective cell migration refers to the movement of a group of cells that maintain contact and read guidance cues cooperatively (*Rorth, 2009*). This mechanism has been studied in several contexts, such as wound healing, angiogenesis, and neural crest (NC) migration. However, how cell proliferation impacts collective cell migration, and vice versa, remains largely unknown. The molecular signals that may couple these two fundamental processes remain equally unclear.

The NC is a mesenchymal cell population that arises early in development and migrates throughout the body, giving rise to a variety of cell types (neurons, glia, pigment cells, etc.). The NC's stereotypical migratory behaviour (*Gammill and Roffers-Agarwal, 2010*) and similarity to metastatic cells (*Maguire et al., 2015*) make this cell type an ideal model to study the mechanisms of collective cell migration in vivo. Our previous work has shown that zebrafish trunk neural crest (TNC) migrate collectively forming single-file chains (*Richardson et al., 2016*). One cell at the front of the chain, the leader, is the only cell capable of instructing directionality to the group, while follower cells trail the leader. This division of roles into leaders and followers has been observed in other collectively migrating systems (*Theveneau and Linker, 2017*). Moreover, histopathological studies from cancer samples and cell lines show clear morphological and molecular differences between the invasive front, leaders, and the lagging cells, followers (*Pandya et al., 2017*). One outstanding question from these studies is what are the signals that determine leader versus follower migratory identities?

Notch signalling is a cell-cell communication pathway that directly translates receptor activation at the membrane into gene expression changes. Notch receptors are activated by membrane-bound ligands of the Delta/Serrate/Lag2 family. Upon ligand binding, Notch receptors are cleaved by γ-secretases releasing their intracellular domain (NICD). Subsequently, NICD translocates to the nucleus, binds the CBF1/Su(H)/Lag-1 complex, and initiates transcription (*Bray, 2016*). Among the direct Notch targets are members of the Hes gene family, which encode transcriptional repressors able to antagonise the expression of specific cell fate determinants and Notch ligands, generating a negative feedback loop in which cells with high Notch receptor activity downregulate the expression of Notch ligands, and cannot activate the pathway in their neighbours. Hence, adjacent cells interacting through the Notch pathway typically end up with either low or high levels of Notch activity and adopt distinct fates, a mechanism known as lateral inhibition (*Lewis, 1998*). Interestingly, Notch signalling has also been implicated in cell migration (*Giniger, 1998*; *Leslie et al., 2007*; *Timmerman et al., 2004*) and promotes invasive behaviours during cancer progression (*Reichrath and Reichrath, 2012*). Furthermore, lateral inhibition is implicated in the allocation of migratory identities during angiogenesis (*Phng and Gerhardt, 2009*), trachea formation in *Drosophila* (*Caussinus et al., 2008*), and in cell culture (*Riahi et al., 2015*). Whether Notch signalling plays a similar role in the context of mesenchymal cell migration is unknown. Notch signalling is required for NC induction (*Cornell and Eisen, 2005*), and its components and activity remain present in migrating NC (*Liu et al., 2015*; *Rios et al., 2011*). Nevertheless, the role of Notch during NC migration remains unclear. Cardiac NC are reported to develop normally under lack of Notch signalling (*High et al., 2007*). However, using different genetic tools, it has been shown that both gain and loss of Notch function led to the lack of NC derivatives (*Mead and Yutzey, 2012*). Moreover, in *Xenopus* the loss of Notch effectors leads to aberrant NC migration (*Vega-López et al., 2015*).

The Notch pathway has not only been implicated in cell fate allocation, but it is also important for cell proliferation. Depending on the context, Notch can inhibit or promote cell cycle progression (*Campos et al., 2002*; *Carlson et al., 2008*; *Devgan et al., 2005*; *Fang et al., 2017*; *Georgia et al., 2006*; *Mammucari et al., 2005*; *Nguyen et al., 2006*; *Nicoli et al., 2012*; *Noseda et al., 2004*; *Ohnuma et al., 1999*; *Park et al., 2005*; *Patel et al., 2016*; *Rangarajan et al., 2001*; *Riccio et al., 2008*; *Zalc et al., 2014*). Indeed, Notch target genes include important cell cycle regulators such as CyclinD1, p21 and MYC (*Campa et al., 2008*; *Guo et al., 2009*; *Joshi et al., 2009*; *Palomero et al., 2006*; *Ronchini and Capobianco, 2001*).

Using a combination of in vivo and in silico approaches, we have established that differences in Notch activity between premigratory TNC select the leader cell. Cells with high levels of Notch signalling adopt a leader identity, while cells that lack Notch activity become followers. Our data show that a single progenitor cell in the premigratory area divides asymmetrically, giving rise to a large prospective leader and smaller follower cell. We propose that this original small asymmetry generates

differences in Notch activity between TNC that are thereafter enhanced by cell-cell communication through Notch lateral inhibition. Differences in Notch activity in turn drive distinct cell cycle progression patterns and regulate the expression of *phox2bb*. Leader cells undergo the $G_1/S$ transition faster and remain in S-phase for longer than follower cells. Moreover, continuous progression through the cell cycle is required for TNC migration. Taken together, our results support a model in which the interaction between Notch and the cell cycle defines leader and follower migratory behaviours.

## Results

### Notch signalling is required for TNC migration

NC cells are induced at the border of the neural plate early during development. The prospective NC expresses Notch components, and Notch activity is required for NC induction (*Cornell and Eisen, 2005*). Our analysis reveals that Notch components remain expressed in NC after induction, suggesting that Notch signalling may also be involved in later aspects of NC development (*Figure 1—figure supplement 1*). Moreover, analysis of the Notch activity reporter line 12xNRE:egpf (*Moro et al., 2013*) shows that Notch signalling levels vary widely between premigratory TNC (*Figure 1*), suggesting that Notch may play a role after TNC induction. To explore the role of Notch in TNC development, we first aimed to define the stage at which NC induction becomes independent of Notch signalling. To this end, we treated embryos with the γ-secretase inhibitor DAPT (*Richter et al., 2017*) and assessed expression of NC marker. Our results showed that Notch inhibition impairs TNC induction up to 11 hours post-fertilization (hpf; *Figure 2*) and confirmed previous reports that induction of the cranial and vagal NC populations is independent of Notch signalling (*Cornell and Eisen, 2000*). Next, we analysed the effect of Notch inhibition at 12 hpf on the development of TNC derivatives. We found a reduction in all TNC derivatives (neurons, glia, and pigment cells; *Figure 3A–F*) upon Notch inhibition, suggesting that Notch activity is important in a process subsequent to induction, yet prior to differentiation. We next explored whether TNC migration is affected by Notch inhibition. Analysis of *crestin* expression showed a reduction in the number of TNC cell chains formed and in their ventral advance upon DAPT treatment (*Figure 3G–J*), which likely explains the lack of TNC derivatives at later stages. We then asked whether these results are due to a delay or a halt of migration. To this end, embryos were treated with DAPT from 12 hpf for 6–12 hr and processed for *crestin* expression. Decreased numbers of migratory chains were observed at all timepoints, but as embryos developed new chains were formed, indicating that the blockade of Notch signalling delays TNC migration (*Figure 3K*). Comparable results were obtained by inhibiting Notch genetically in embryos where the dominant-negative form of Suppressor of Hairless is under the control of a heat shock element (*Latimer et al., 2005*; hs:dnSu(H); *Figure 3L*). We reasoned that if Notch inhibition delays the onset of TNC migration, its overactivation might lead to TNC migrating earlier, leading to an increased number of chains. To test this, we induced NICD expression in all tissues by heat shock of hs:Gal4;UAS:NICD embryos (*Scheer and Campos-Ortega, 1999*). To our surprise, Notch gain of function (GOF) and loss of function (LOF) resulted in almost identical phenotypes, both showing a similar reduction of TNC chain numbers (*Figure 3L*). Taken together, these results show that precise regulation of Notch signalling levels is required for TNC migration.

### In vivo Notch activity allocates TNC migratory identity

Interestingly, Notch signalling is required during collective migration to define distinct identities (*Phng and Gerhardt, 2009*; *Caussinus et al., 2008*; *Riahi et al., 2015*). To test whether Notch plays a similar role in TNC migration, we performed live-imaging analysis of TNC migration under lack (inhibition and LOF) or overactivation (GOF) of Notch signalling (*Figure 4*, *Figure 4—videos 1 and 2*). Our previous work defined a leader as the cell that retains the front position of the chain throughout migration, advancing faster and in a more directional manner than followers (*Richardson et al., 2016*). Under Notch inhibition (treatment with γ-secretase inhibitor Compound E; *Richter et al., 2017*), TNC remain motile with a single cell initiating the movement of the chain, but in contrast to control treatment (DMSO) the leader cell is unable to retain the front position and is overtaken by one or several followers (*Figure 4A and C*, *Figure 5A and B*, *Figure 4—video 1*). The overtaking follower cell, in turn, is not always able to retain the front position and can be overtaken by cells further behind in the chain. This loss of group coherence corresponds with a reduction in ventral advance, with most leader

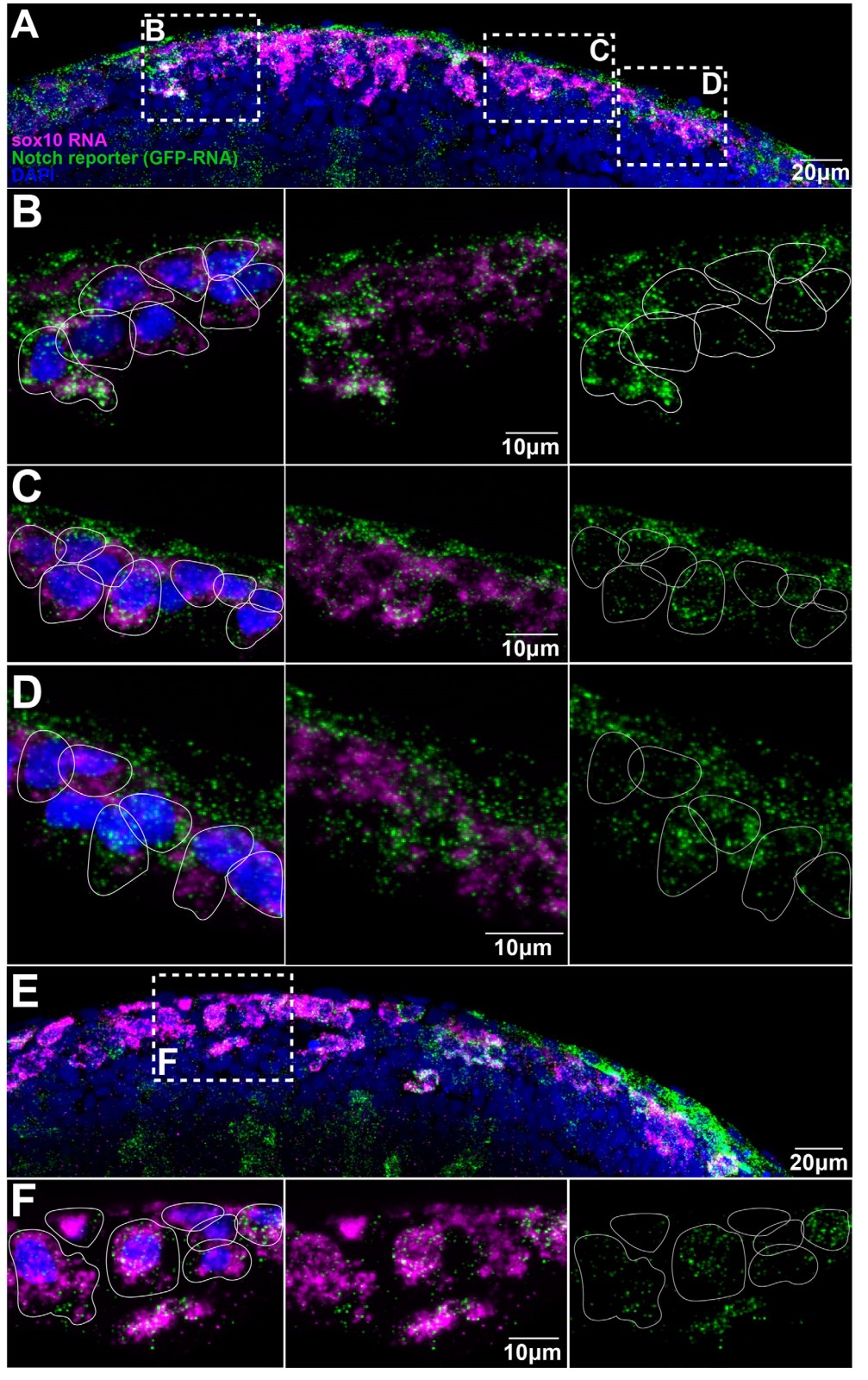

**Figure 1.** Trunk neural crest (TNC) present different levels of Notch activity. (**A, E**) Images of two different Notch reporter 12xNRE:egfp embryos (18 hpf) stained for *sox10* (magenta) and GFP (green) RNAs, and nuclei stained with DAPI (blue). (**B**) Enlargement of the anterior area in (**A**). (**C**) Enlargement of the more posterior area in (**A**).

*Figure 1 continued on next page*

*Figure 1 continued*

(**D**) Enlargement of the anterior most posterior area in (**A**). (**F**) Enlargement of the outlined area in (**E**). Anterior to the left, dorsal top. White lines show approximate cell boundaries.

The online version of this article includes the following figure supplement(s) for figure 1:

**Figure supplement 1.** Expression of Notch signalling components during trunk neural crest (TNC) migration.

cells unable to move beyond the neural tube/notochord boundary (NT/not; *Figure 4C*, *Figure 5A and C*). This behaviour leads to an accumulation of cells at the NT/not, where some cells repolarise moving anterior or posteriorly and crossing the somite boundary and, in some cases, joining adjacent chains. Analysis of single-cell tracking showed that under Notch inhibition leader cells also have decreased speed and directionality (*Figure 5D and E*). Similar results were observed when Notch inhibition was achieved genetically by driving overexpression of dnSu(H) through heat shock in the entire embryo (not shown; hs:dnSu(H) line). Together, these results strongly suggest that upon lack of Notch signalling the TNC population is formed solely by follower cells that are unable to coordinate the movement of the group. Nevertheless, Notch signalling is important for the development of tissues surrounding TNC that act as a substrate for migration, raising the possibility that Notch signalling does not act cell-autonomously in TNC and instead the phenotypes observed are simply the consequence of somite and/or neural tube malformations. However, this appears unlikely as somite development (formation, patterning, and differentiation) and neuron formation are not affected by Notch inhibition at the axial level analysed (*Figure 4—figure supplement 1*). Next, we directly tested whether Notch signalling is autonomously required in TNC by inhibiting Notch activity exclusively in NC at the time of migration. To this end, we generated a new UAS:dnSu(H) line and crossed it with Sox10:Kalt4 fish (*Alhashem et al., 2021*). In the resultant embryos, all NC express Gal4 fused to the oestrogen receptor binding region (Gal4-ER) and are fluorescently labelled by nuclear-RFP. Under normal conditions, Gal4-ER is maintained inactive in the cytoplasm, whilst upon addition of tamoxifen, Gal4-ER is translocated to the nucleus activating transcription from the UAS:dnSu(H) transgene (*Figure 4—figure supplement 2*). We found that autonomous inhibition of Notch signalling in NC phenocopies the chemical inhibition. Leader cells are unable to retain the front position, being overtaken by followers, and ventral advance is reduced with cells accumulating at the NT/not boundary (*Figure 4D*, *Figure 5A–C*, *Figure 4—video 2*). Moreover, leader cells adopt followers' migratory parameters, showing decreased speed and directionality (*Figure 5D and E*), confirming that Notch activity is autonomously required in TNC for identity allocation, and suggest that in the absence of Notch signalling a homogenous group of followers is established. In view of these results, we hypothesised that a homogeneous group of leaders would be formed upon Notch overactivation. Using a similar strategy, Notch overactivation was induced in the whole embryo (not shown, hs:Gal4;UAS:NICD; *Scheer and Campos-Ortega, 1999*), or exclusively in NC (Sox10:Kalt4;UAS:NICD), and migration was analysed by live imaging. Similar results were obtained in both experimental conditions: group coherence is lost, leader cells are overtaken by followers, and ventral advance is impaired (*Figure 4F*, *Figure 5A–C*, *Figure 4—video 2*). Interestingly, in Notch GOF conditions follower cells adopt leaders' characteristics, moving with increased speed, but all cells in the chain follow less directional trajectories, which hinders the ventral advance of the group (*Figure 5D and E*), indicating that all cells in the chain migrate as leaders. Next, we tested whether the behavioural changes observed upon Notch alterations were mirrored by molecular changes by using the leader marker *phox2bb*. In control conditions, *phox2bb* transcripts are highly enriched in the leader cells from early stages of migration (*Figure 6A, B, and G*; *Alhashem et al., 2022a*). Consistent with expectations, upon Notch overactivation *phox2bb* is expressed by all the cells in the chain (*Figure 6C, D, and G*), while its expression is absent when Notch is inhibited (*Figure 6E–G*). These data show that Notch activity controls *phox2bb* expression and allocates TNC migratory identity.

In summary, our in vivo and molecular data show that Notch signalling is required autonomously in TNC for migratory identity allocation. TNC with high levels of Notch express *phox2bb* and become leaders, while cells with low Notch activity migrate as followers. Alterations of Notch signalling lead to a homogeneous TNC group with a single migratory identity that is unable to undergo collective migration. Taken together, these data suggest Notch lateral inhibition as the mechanism responsible for TNC migratory identity acquisition.

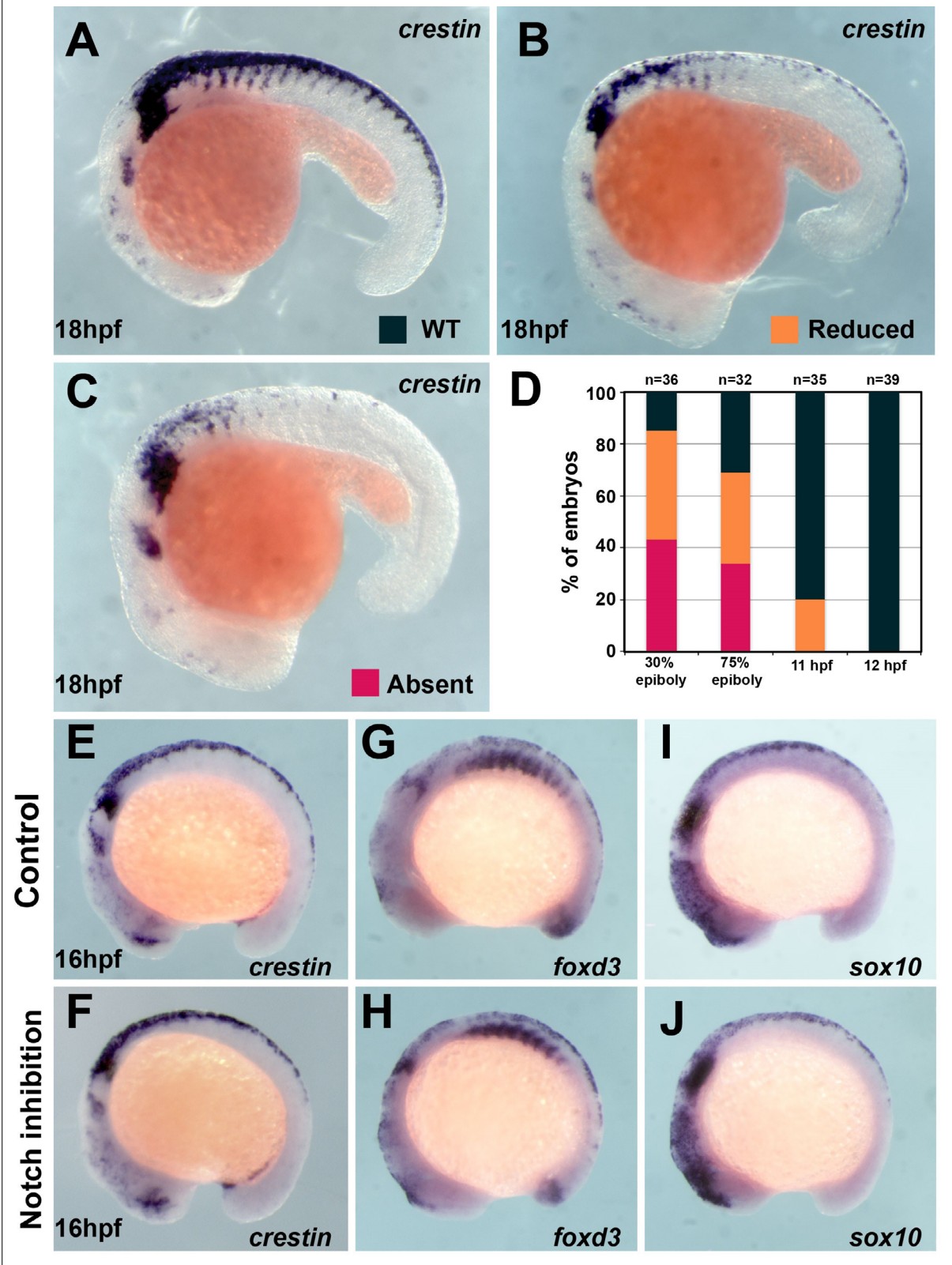

**Figure 2.** Trunk neural crest (TNC) induction is independent of Notch signalling after 12 hpf. (**A**) *crestin* in situ hybridisation in wildtype (WT) embryo at 18 hpf. (**B, C**) *crestin* in situ hybridisation in DAPT-treated embryos: (**B**) reduced or (**C**) absent TNC. (**D**) Quantification of the *crestin* expression phenotypes upon DAPT treatment (phenotypes: WT, black; reduced, orange; absent, red; 30% epiboly n = 38, 75% epiboly n = 32, 11 hpf n = 35, 12 hpf n = 39). (**E–J**) In situ hybridisation for neural crest (NC) markers in representative control (DMSO) and DAPT-treated embryos from 12 to 16 hpf. (**E,**

*Figure 2 continued on next page*

*Figure 2 continued*

**F**) *crestin* (DMSO n = 32, DAPT n = 38), (**G, H**) *foxd3* (DMSO n = 16, DAPT n = 35), and (**I, J**) *sox10* (DMSO n = 27, DAPT n = 29). Anterior to the left, dorsal top.

## In silico modelling predicts that more than one leader is required for TNC migration

Our in vivo analysis shows that upon both Notch inhibition and overactivation TNC are unable to undergo collective migration due to lack of group coherence. On the other hand, our molecular analysis shows that upon Notch inhibition an all-followers group is established, while Notch overactivation leads to the formation of an all-leaders group. To gain a better understanding of these paradoxical results, we took an in silico approach, developing a discrete element model of TNC migration. Cells were simulated as 2D particles moving into a constrained space and endowed with intrinsic motility. Four variables control cell movement in the model: contact inhibition of locomotion (CIL) and co-attraction (co-A) define movement directionality and group cohesion, while volume exclusion regulates cell overlap, intuitively understood as cell size, while a noise element (zeta) was added to the cell's trajectory (*Figure 7A*). A multi-objective scoring system, based on in vivo measurements, was developed to evaluate how close simulations with different underlying mechanisms matched chain behaviours. The scores were (1) chain cohesion, a maximum distance of 57 µm is allowed between adjacent cells; (2) single file migration for at least 80% of the simulation; (3) followers undergo rearrangements, while (4) leaders retain the front position, and (5) the chain should advance to the end of the migratory path (*Figure 7B*). Using this analysis and a parsimonious modelling approach, we attempted to match in vivo TNC migration with the simplest form of the model, only adding complexity incrementally in an effort to find the minimal set of predicted mechanisms required. We first simulated chains composed of homogeneous cells and systematically covaried all parameters. We found no parameter combination able to match all scores, confirming our previous findings that cell heterogeneity is required for TNC migration (*Figure 7C*; *Richardson et al., 2016*). Evidence from other systems (*Astin et al., 2010*; *Bentley et al., 2014*; *Parkinson and Edwards, 1978*; *Theveneau and Mayor, 2013*) led us to hypothesise that differences in the CIL response between cells may be at play. Thus, we simulated chains in which only cells of different identities present CIL (Diff CIL; *Figure 7A*). These simulations match several scores, but chains are unable to reach the end of the migratory path (*Figure 7C*, *Figure 4—video 3*). Next, we varied Diff CIL intensity, co-A, and cell size (volume exclusion) for leader cells. Interestingly, the model is only able to recapitulate control conditions when the difference between leaders and follower is maximal for all variables. Nevertheless, it is unable to recapitulate Notch GOF and LOF phenotypes (*Figure 7C*). Our previous results show that differences in Notch signalling establish migratory identities, suggesting that lateral inhibition may be the mechanism at play. To explore whether different outcomes of lateral inhibition may allow the model to simulate Notch altered conditions (GOF and LOF), different ratios of leader/follower cells were simulated. We first tested a 1:1 ratio, surprisingly this chain architecture over-migrates, moving beyond the end of the pathway (*Figure 7C*, *Figure 4—video 3*). Interestingly, we found that several parameter combinations from the 1:2 and 1:3 leader/follower ratios were able to recapitulate in vivo control condition, as well as the loss of group coherence and ventral advance observed in Notch GOF (all leader simulation) and LOF (all follower simulation; *Figure 4B, E, and G*, *Figure 5*, *Figure 4—video 3*). In these simulations, the six parameter combinations that match all in vivo scores had followers at the low setting, while leaders' CIL intensity took medium or high values, cell size took medium or low values, and co-attraction took all levels. Nevertheless, all these parameter combinations endow the leader with enhanced migratory behaviour.

Next, we used a linear discriminant analysis (LDA) to study which of the model parameters bear most weight in the definition of leader and follower identity. LDA is a dimensionality reduction method that projects the data onto a lower dimensional space minimizing the variation within classes (e.g. between leaders) and maximizing the variation between classes (leaders versus followers), allowing the hierarchical ordering of the factors that best explain the class separation. First, we used the in vivo data to determine whether leaders and followers were properly separated by LDA. A visual inspection of the data makes clear that LDA works well to classify migratory identities (*Figure 7D*). Moreover, the LDA shows that ventral distance is the most important variable separating leaders from followers, with

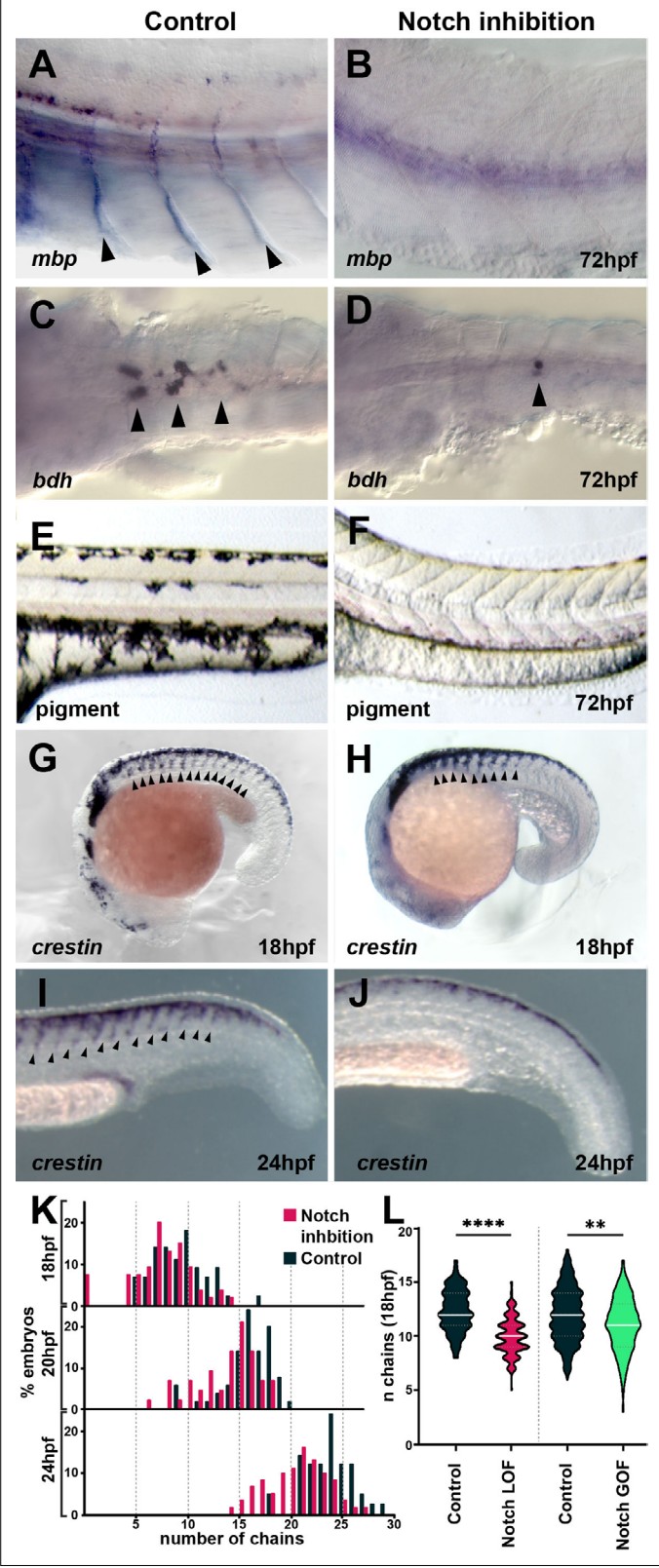

**Figure 3.** Notch signalling is required for trunk neural crest (TNC) migration and derivatives formation. (**A, B**) *Glial marker mbp* in situ hybridisation upon (**A**) control (DMSO; n = 15) and (**B**) DAPT (n = 20) treatment from 12 hpf. (**C, D**) *Neuronal marker bdh* in situ hybridisation upon (**C**) control (DMSO; n = 25) and (**D**) DAPT (n = 18) treatment from 12 hpf. (**E, F**) Pigmentation upon (**E**) control (DMSO; n = 40) and (**F**) DAPT (n = 52) treatment from

*Figure 3 continued on next page*

*Figure 3 continued*

12 hpf. (**G, H**) Neural crest marker *crestin* in situ hybridisation upon (**G**) control (DMSO) and (**H**) DAPT treatment from 12 to 18 hpf. (**I, J**) *crestin* in situ hybridisation upon (**I**) control (DMSO) and (**J**) DAPT treatment from 12 to 24 hpf. (**K**) Quantification of migratory chain formation upon control (DMSO) and DAPT treatment from 12 to 18 hpf (DMSO n = 98; DAPT n = 126), 20 hpf (DMSO n = 111; DAPT n = 109), and 24 hpf (DMSO n = 42; DAPT n = 61). (**L**) Quantification of migratory chain formation in control (HS:Gal4; n = 516), Notch loss of function (LOF) (HS:dnSu(H); n = 220), and gain of function (GOF) conditions (HS:Gal4xUAS:NICD; n = 142) heat shocked at 11 hpf and analysed at 18 hpf. Mann–Whitney *U*-test, control vs. LOF ****p<0.0001, control vs. GOF **p=0.0020. Anterior to the left, dorsal top, except in (**C, D**) anterior left, ventral view. Arrowheads indicate gene expression. All treatments performed from 12 hpf.

speed and directionality playing a less dominant role (*Figure 7E*). Next, we used this method to assess the importance of each of the model parameters. CIL intensity appears to be the parameter that most differ between leader and follower cells, while heterogeneity in the other parameters is not essential (*Figure 7F*). Taken together, the in silico data confirms our previous conclusion that TNC chains are a heterogeneous group. Remarkably, it also predicts CIL intensity to be the most important distinction between leaders and followers. Finally, the model anticipates that TNC chains are formed of leaders and followers in a 1:2 or 1:3 ratio.

## Leader cells arise from the asymmetric division of a progenitor cell

Cell size is a prominent characteristic distinguishing leader from follower cells. Leaders are almost twice as big as followers during migration and this difference is evident before migration initiation (*Richardson et al., 2016*), suggesting that size disparity arises at birth or shortly thereafter. Interestingly, differential cell size emerged as an important parameter in our in silico analysis, contributing to more realistic leader/follower coordination behaviours. To understand the origin of these size differences, we investigated whether leader and follower cells share a common progenitor, and at which point differences in size become apparent. To this end, we imaged FoxD3:mCherry;H2aFVA:H2a-GFP embryos. The FoxD3:mCherry reporter (*Hochgreb-Hägele and Bronner, 2013*; *Lukoseviciute et al., 2018*) labels NC from early stages and allows us to define TNC identity at later stages by their migratory position. Moreover, the nuclear marker H2aFVA:H2a-GFP (*Pauls et al., 2001*) was used to track single cells and their divisions. Tracking analysis shows that the asymmetric division of a single progenitor cell in each body segment gives rise to a larger cell that becomes a leader (102 ± 20 μm²), and a smaller sibling that migrates as follower (72 ± 9 μm²; *Figure 8A and B*, *Figure 8—video 1*). In contrast, all other progenitors divide symmetrically, giving rise to two follower cells (87 ± 27 μm²; *Figure 8C and D*). We also noticed that the leader progenitors' divisions are spatially restricted to the anterior quarter of the premigratory area in each segment, while the followers' progenitor divisions take place across the premigratory area (*Figure 8E*).

We then reasoned that leader cells, being bigger, may undergo the next division in a shorter time span than follower cells, and in consequence, mitotic figures would be observed at different, but consistent, positions in their trajectory. Indeed, we found two different patterns of divisions in respect to migration: (1) cells that first *D*ivide and then *M*igrate (D→M) or (2) cells that first *M*igrate and then *D*ivide (M→D; *Figure 8F and G*, *Figure 8—video 2*). Interestingly, we found that the patterns of cell division correlate with cell identity. Most leader cells divide during migration (M→D: 86%), while the bulk of follower cells divide before migration initiation (D→M: 90%, *Figure 8H*). These patterns result in leader and follower cells dividing at distinct positions, 74% of leaders divide at the NT/not boundary (65.3 ± 9.6 μm), while 85% of followers divide mostly within the premigratory area or in the dorsal-most region of the somite (42 ± 12.4 μm; *Figure 8I*). Together, these results show that leader cells arise from the asymmetric division of a progenitor. Thereafter, leader and follower cells show distinct locations and patterns of division, suggesting that leaders and followers progress asynchronously through the cell cycle, which may influence their migratory behaviour.

## Cell cycle progression is required for TNC migration

To test the role of cell cycle progression in TNC migration directly, we used inhibitory drugs. The S-phase inhibitor aphidicolin blocks over 94.7% ± 4.5% of mitotic figures after 3 hr of treatment, while the G2/M inhibitor genistein prevents 90% ± 10% of divisions within 6 hr, but neither treatment affects

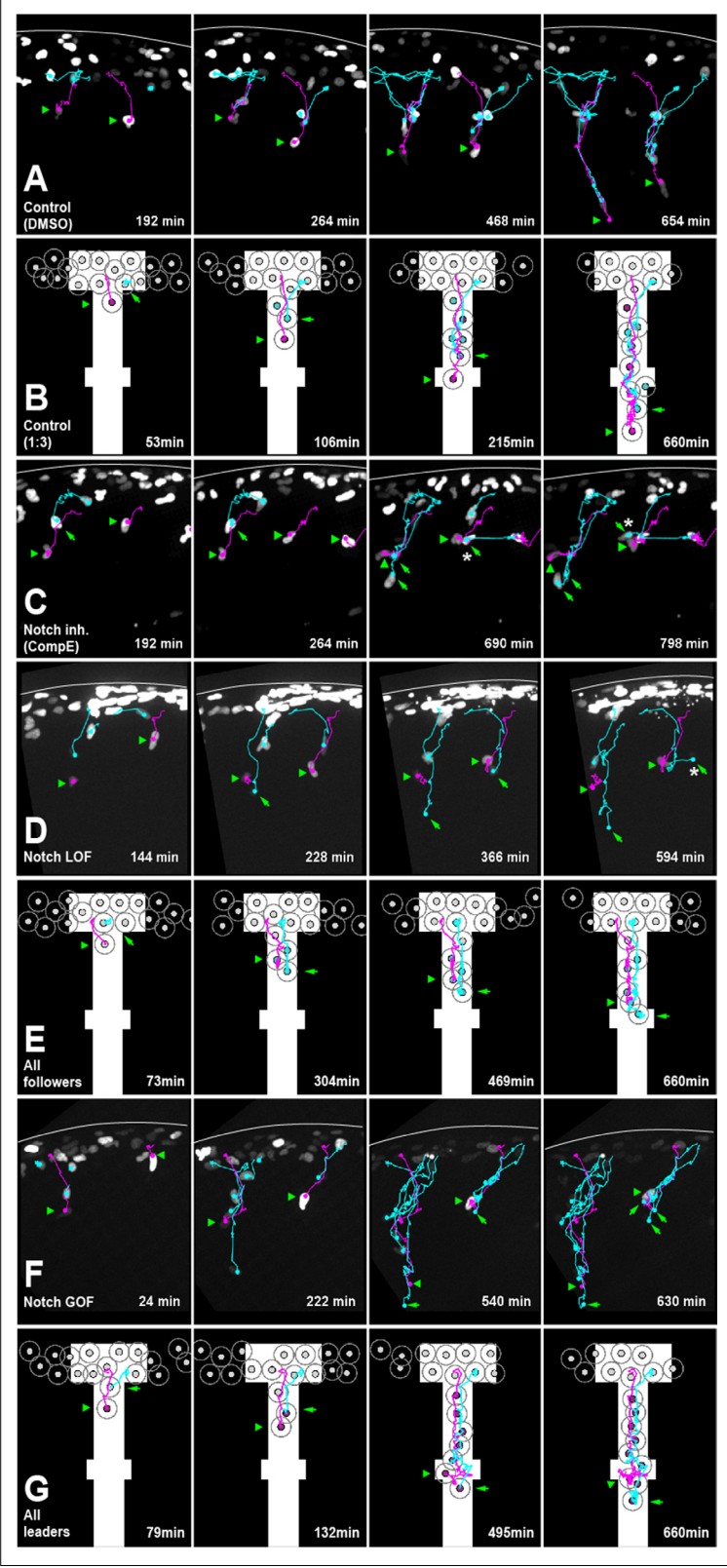

**Figure 4.** Notch activity allocates trunk neural crest (TNC) migratory identity. (**A**) Selected frames from in vivo imaging of Sox10:Kalt4 control (DMSO treated) embryos. (**B**) Selected frames from control simulation with 1:3 leader/follower ratio. (**C**) Selected frames from in vivo imaging under Notch-inhibited condition, Sox10:Kalt4 embryos treated with CompE. (**D**) Selected frames from in vivo imaging of Notch loss of function (LOF) condition,

*Figure 4 continued on next page*

*Figure 4 continued*

Sox10:Kalt4; UAS:dnSu(H) embryos. (**E**) Selected frames from all followers simulation. (**F**) Selected frames from in vivo imaging of Notch gain of function (GOF) condition Sox10:Kalt4; UAS:NICD embryos. (**G**) Selected frames from all leaders simulation. Magenta tracks and green arrowheads indicate leaders; green arrows and cyan tracks follower cells. Asterisks indicate cells crossing somite borders. White line marks dorsal midline. Anterior to the left, dorsal up. Time in minutes.

The online version of this article includes the following video and figure supplement(s) for figure 4:

**Figure supplement 1.** Somites and neural tissue formation are not altered by Notch inhibition.

**Figure supplement 2.** UAS:dnSu(H) transgenic line.

**Figure 4—video 1.** Notch inhibition disrupts trunk neural crest (TNC) migratory identity allocation.
https://elifesciences.org/articles/73550/figures#fig4video1

**Figure 4—video 2.** Notch gain and loss of function disrupts trunk neural crest (TNC) migratory identity allocation.
https://elifesciences.org/articles/73550/figures#fig4video2

**Figure 4—video 3.** In silico simulation of trunk neural crest (TNC) chain migration.
https://elifesciences.org/articles/73550/figures#fig4video3

---

NC induction (*Figure 9—figure supplement 1*). Inhibition of cell cycle progression by either of the treatments resulted in reduced numbers of migratory chains and decreased ventral advance (control 19 ± 2, genistein 10 ± 3, aphidicolin 6 ± 2 chains; *Figure 9A–H*). This result was not due to the loss of cell motility as premigratory TNC cells actively extend protrusions and move along the anteroposterior axis but are unable to migrate ventrally (*Figure 9—video 1*). Importantly, these effects were not a consequence of cell death or the permanent impairment of motility as TNC reinitiate migration and form new chains upon drug withdrawal (*Figure 9G and H*), showing that active cell cycle progression is required for migration. Next, we directly analysed TNC cell cycle progression in vivo. To this end, we imaged Sox10:FUCCI embryos (*Rajan and Gallik, 2018*), in which TNC nuclei are RFP-labelled during $G_1$ and GFP-labelled during S and $G_2$. Tracking analysis shows differential cell cycle progression, with most leader cells initiating migration in S-phase (79%), while followers start movement during $G_1$ (77%; *Figure 9I and J*, *Figure 9—video 2*). These results show that cell cycle progression is required for migration and that leader and follower cells initiate movement at different points of the cell cycle, suggesting an intimate connection between cell growth and movement.

## Leader and follower cells progress through the cell cycle at different rates

Next, we studied TNC cell cycle progression in detail. First, we asked whether leaders and followers differ in the total length of their cell cycle. Measurements of the time span between two consecutive mitoses showed no significant differences in the total length of the cell cycle between leaders and followers (13.6 ± 1.2 and 13.3 ± 1.4 hr, respectively; *Figure 10B*). Next, we examined the length of each phase of the cell cycle by imaging the characteristic nuclear labelling pattern of the PCNA-GFP fusion protein (*Leung et al., 2011*). Sox10:Kalt4 embryos, in which all NC can be recognised by nuclear RFP expression, were injected with PCNA-GFP mRNA and live imaging was performed. PCNA-GFP shows uniform nuclear GFP labelling during $G_1$, intense fluorescent nuclear puncta characterise the S-phase, these puncta dissipate during $G_2$ restoring homogeneous nuclear fluorescence, at the onset of mitosis PCNA is degraded and TNC are recognised solely by nuclear RFP (*Figure 10A*, *Figure 10—video 1*). In these embryos, leader cells initiate migration during S-phase and followers in $G_1$, confirming our FUCCI results and establishing that PCNA overexpression does not introduce artefacts to cell cycle progression (*Figure 10—figure supplement 1*). Using this tool, we measured the length of the cell cycle phases in TNC. We found striking differences in the time spent in $G_1$- and S-phase between leader and follower cells. Leaders present a short $G_1$ (3.2 ± 0.6 hr) but remain for twice as long in S-phase (8.7 ± 1.3 hr). Followers, on the other hand, present the opposite distribution, remaining for twice as long in $G_1$ (7.4 ± 2.7 hr) than in S-phase (4.6 ± 2.8 hr; *Figure 10C and D*). No significant differences were observed in the length of $G_2$ (leaders 1.6 ± 0.4 hr; followers 1.5 ± 0.3 hr) or M (leaders 0.6 ± 0.1 hr; followers 0.5 ± 0.1 hr). These data show that leader and follower cells present marked differences in the length of $G_1$- and S-phase, suggesting that cell cycle progression may regulate their migratory behaviour.

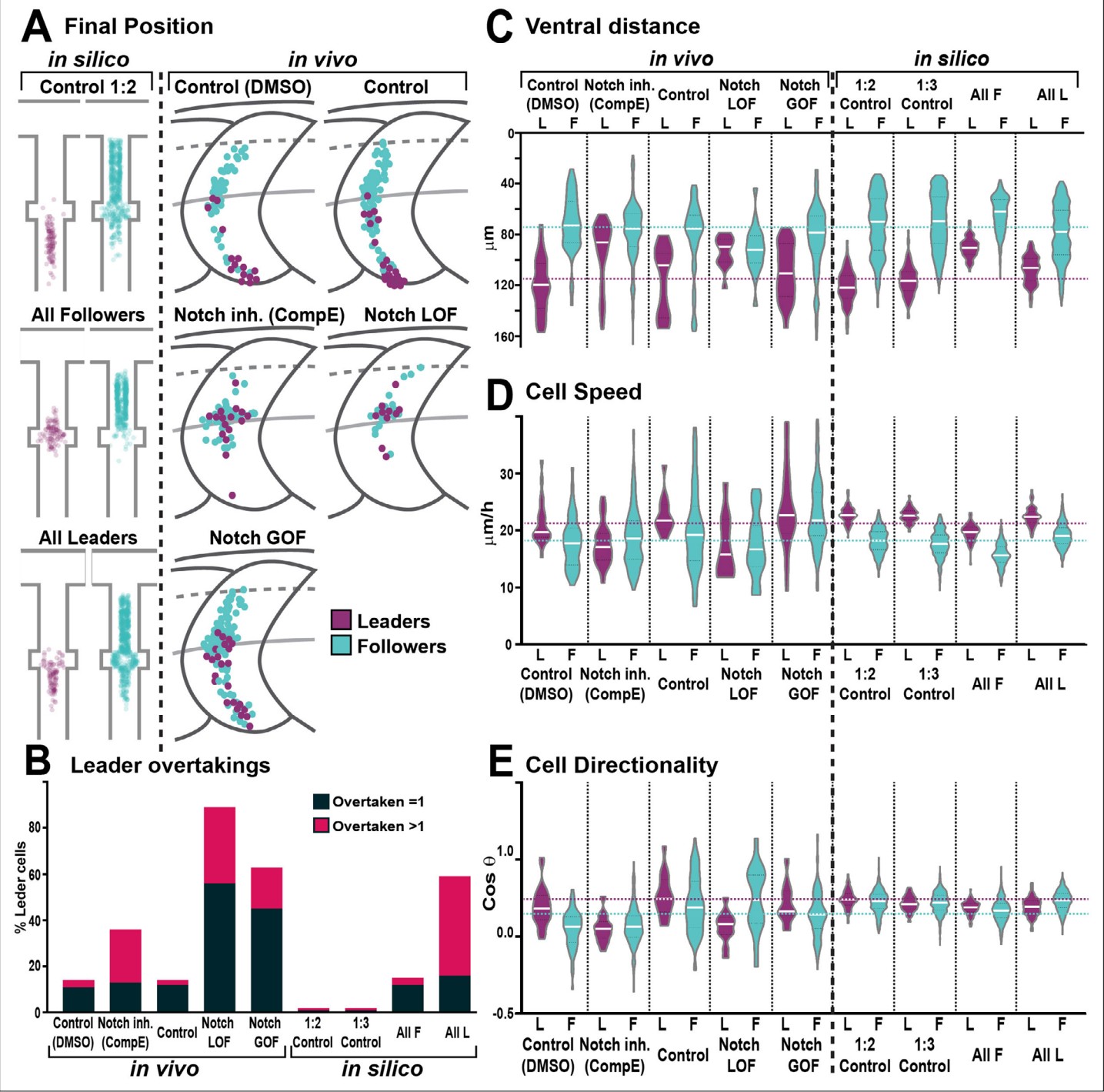

**Figure 5.** Trunk neural crest (TNC) migration measurements in vivo and in silico. (**A**) Final position of each cell in model simulations and in vivo experiments under different conditions. In silico results depicted in confined pathway, in vivo data graphed in model embryo, somites contour and dorsal midline (dark grey lines), edge of the premigratory area (dashed lines), and NT/not boundary (light grey lines). Anterior left, dorsal up. (**B**) Quantification of leader overtaking events in vivo and in silico. Leader overtaken by a single follower is overtaken = 1; leader overtaken by more than one follower cell is overtaken >1. (**C**) Quantification of the ventral advance of cells in vivo and in silico. (**D**) Quantification of cell speed in vivo and in silico. (**E**) Quantification of cell directionality in vivo and in silico. Leader cells in magenta, followers in cyan. Magenta and cyan dashed lines indicate the average values for leaders and followers respectively. Full statistical analysis in *Supplementary file 1*.

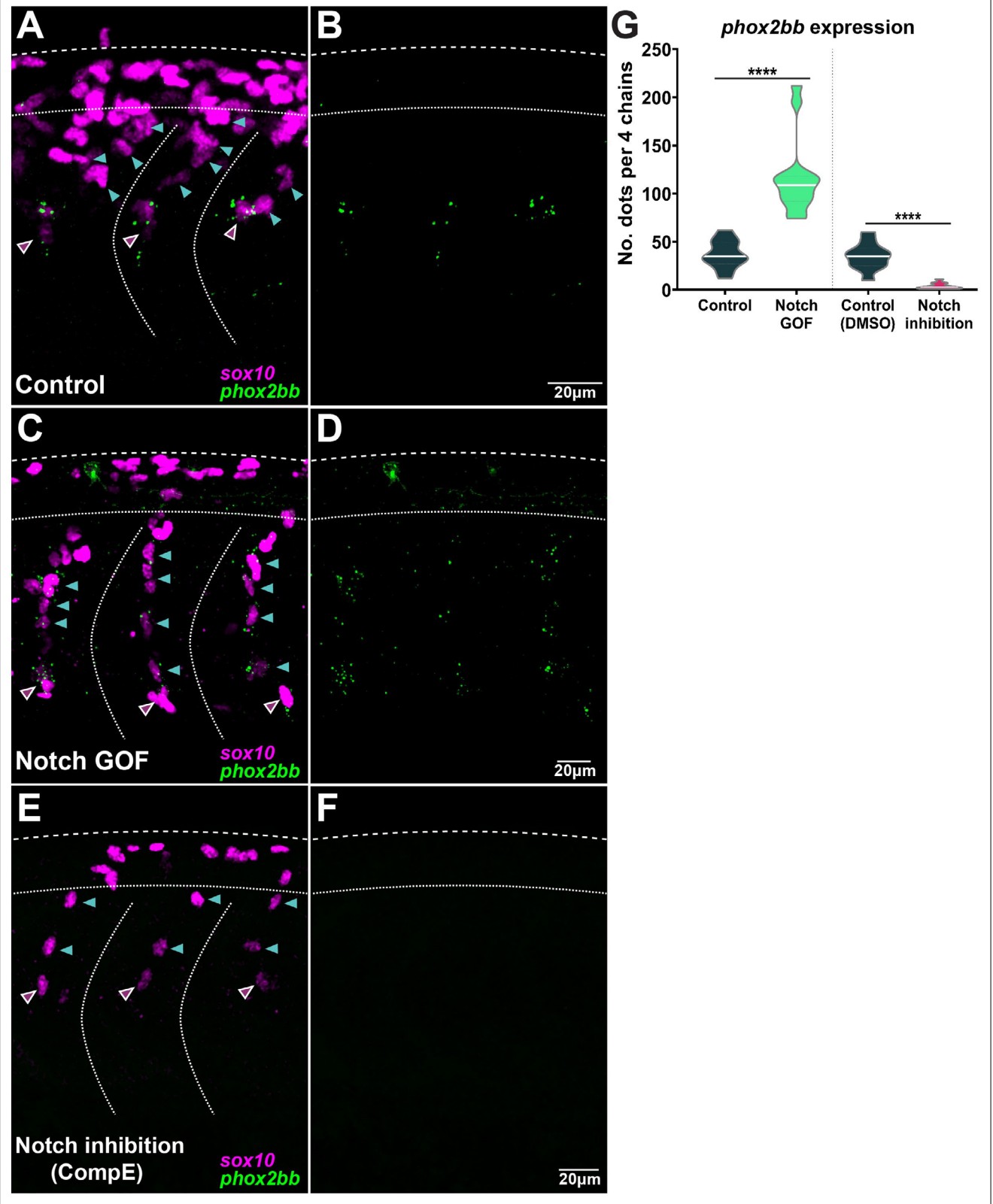

**Figure 6.** Notch signalling controls *phox2bb* expression defining leader cells. (**A, B**) Images of *phox2bb* expression in control embryos (Sox10:Kalt4). (**C, D**) Images of *phox2bb* expression under Notch gain of function (GOF) conditions (Sox10:Kalt4; UAS:NICD embryos). (**E, F**) Images of *phox2bb* expression in Notch inhibition conditions (Compound E). Magenta and cyan arrowheads indicate leaders and followers respectively. (**G**) Quantification of *phox2bb* expression in control (n = 13), Notch GOF (n = 14), and Notch inhibition conditions (n = 11). Welch's *t*-test, Kalt4 control vs. GOF ****p<0.0001, DMSO control vs. inhibition ****p<0.0001.

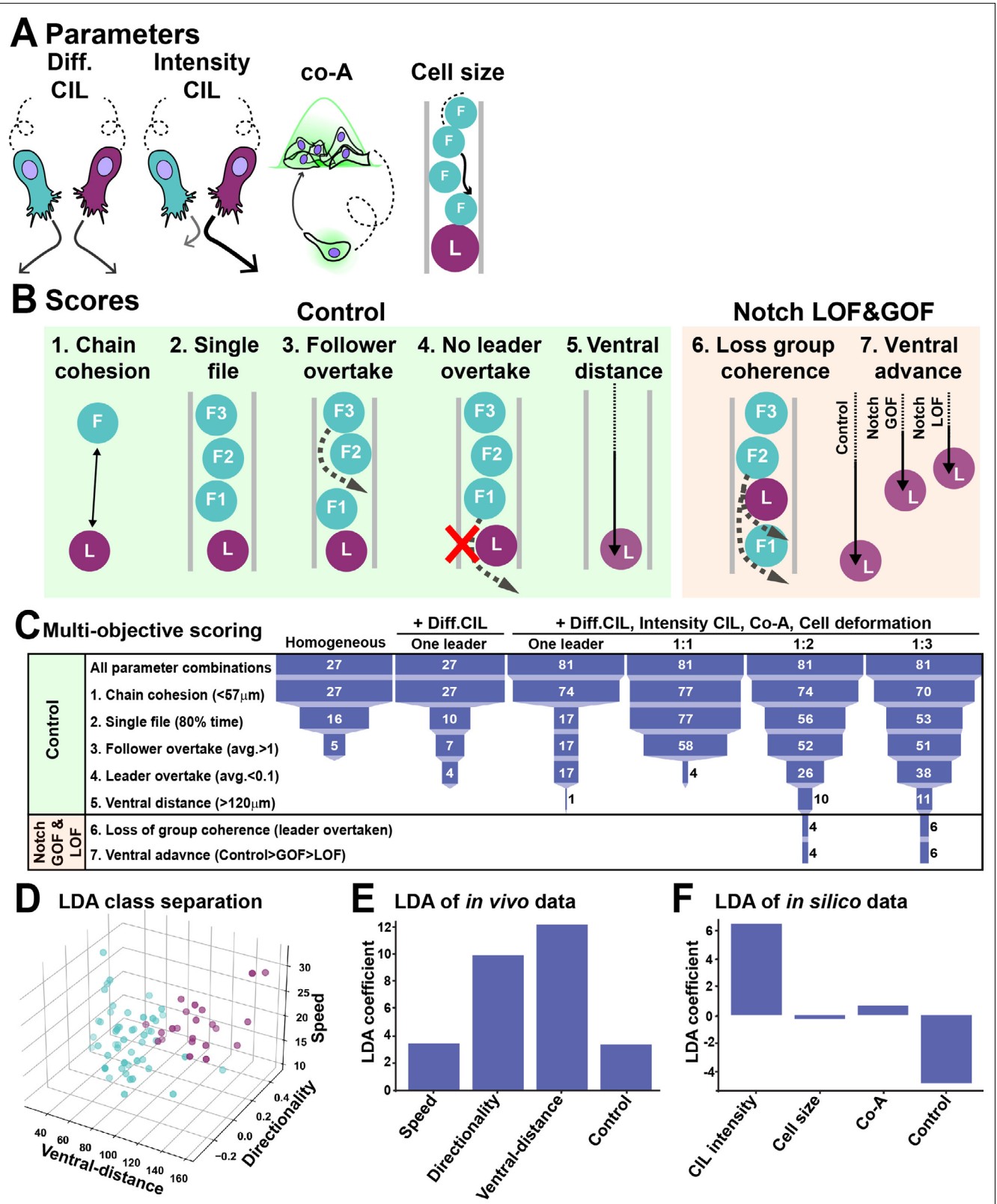

**Figure 7.** In silico modelling of trunk neural crest (TNC) migration. (**A**) Schematics of model parameters. Diff CIL: only leader/follower collisions induce repulsion and change of directionality. Intensity CIL: the leader's response upon collision is stronger than the follower's response. Co-A: co-attraction pulls together cells at a distance. Cell size: volume exclusion. (**B**) Schematics of simulations multi-objective scores. (**C**) Depiction of parameter space analysis showing the number of parameters sets that fulfilled each score when different variables were tested. One leader refers to chains with a single

*Figure 7 continued on next page*

*Figure 7 continued*

leader cell. 1:1, 1:2, and 1:3 refer to leader/follower ratios. (**D**) 3D plot of linear discriminant analysis (LDA). (**E**) LDA coefficients of in vivo data. A random dataset was used as control. (**F**) LDA coefficients of in silico data. A random dataset was used as control.

## Notch signalling regulates TNC cell cycle progression

Our data show that Notch signalling allocates leader and follower identities, cell cycle progression is necessary for TNC migration, and leader and follower cells progress through the cell cycle at different rates. Does Notch signalling regulate cell cycle progression, thus differentiating leader from follower cells? To investigate this question, we measured the total length of the cell cycle and the length of each phase under control and Notch-inhibited conditions. Neither the total cell cycle length (*Figure 11A*) nor the number of TNC (*Figure 9—figure supplement 1*) were affected by alterations of Notch signalling. Remarkably, we found significant differences in the length of $G_1$- and S-phase upon Notch inhibition. Leader cells lose their characteristic cell cycle progression pattern and behave as followers, with a long $G_1$ and a short S-phase (*Figure 11B*). Furthermore, Notch inhibition abolishes the size difference between migratory leader and follower cells, with all cells presenting the average follower's area (*Figure 11C and D*). These data show that Notch activity defines TNC migratory identity by regulating cell cycle progression, cells with low Notch activity remain for longer in $G_1$ behaving as followers. Interestingly, we noticed that Notch inhibition also changes the cell cycle behaviour of the followers' population. While the followers' average length of cell cycle phases is not altered, the dispersion of this population is significantly reduced, with standard deviations cut almost by half (from 2.7 hr to 1.42 hr for $G_1$ and from 2.8 hr to 1.38 hr for S; *Figure 11B*). This prompted us to analyse the frequency distribution of cell cycle phases length. In control conditions, leader cells show a normal distribution with a single peak for $G_1$- and S-phase, as expected for a homogeneous population. Followers, on the other hand, present a bimodal distribution, with the smaller peak coinciding with that of leader cells, and accounting for 26% of followers in $G_1$- and 31% in S-phase (*Figure 11E and F*). Strikingly, these results fulfil the predictions of our in silico model that best recapitulates TNC migration when chains are composed of leaders and followers in a 1:2 or 1:3 ratio. Furthermore, upon Notch inhibition the bimodal distribution of the follower population is lost, with all cells grouped at the major mean (*Figure 11G and H*). Consistent with these data, closer analysis (at higher magnification) of normal *phox2bb* expression shows increased expression in followers at position 3 in addition to that in leaders (*Figure 11I–N*). Taken all together, our data demonstrate that the levels of Notch activity in TNC allocate migratory identity by controlling cell cycle progression and that migratory chains are formed of one leader cell for every three followers.

## Discussion

Collective migration plays an important role in embryogenesis, wound healing, and cancer. The acquisition of specific migratory identities has proven fundamental to angiogenesis, trachea development in *Drosophila*, and cancer metastasis. TNC migrate collectively, forming chains with a leader cell at the front of the group that direct the migration, while follower cells form the body of the chain that trails the leaders. TNC leader and follower identities are established before migration initiation and remain fixed thereafter (*Richardson et al., 2016*). Herein, we have addressed the mechanism that establishes leader and follower identities and can propose the following model (*Figure 12*): (A) premigratory TNC progenitors arise at the dorsal part of the neural tube. The leader's progenitor divides asymmetrically, giving rise to a large prospective leader cell and a small sibling that migrates as a follower. Other progenitors divide symmetrically, giving rise to follower cells. (B) Interactions via Notch signalling results in the prospective leader cell accumulating higher levels of Notch activity, which induces *phox2bb* expression. (C) The combination of high Notch activity and a larger cell size prompts the prospective leader cell to rapidly undergo the $G_1/S$ transition, entering S-phase and initiating migration earlier than its follower siblings, which are smaller and initiate migration whilst in $G_1$. (D) Premigratory cells that have not been in contact with the prospective leader cell or that have lost contact with it due to its ventral advance maintain communication with surrounding premigratory TNC through Notch and undergo a new round of leader cell selection. This working model of TNC migration is supported by both our experimental data and our in silico modelling, and provides a useful conceptual framework for future studies to build upon.

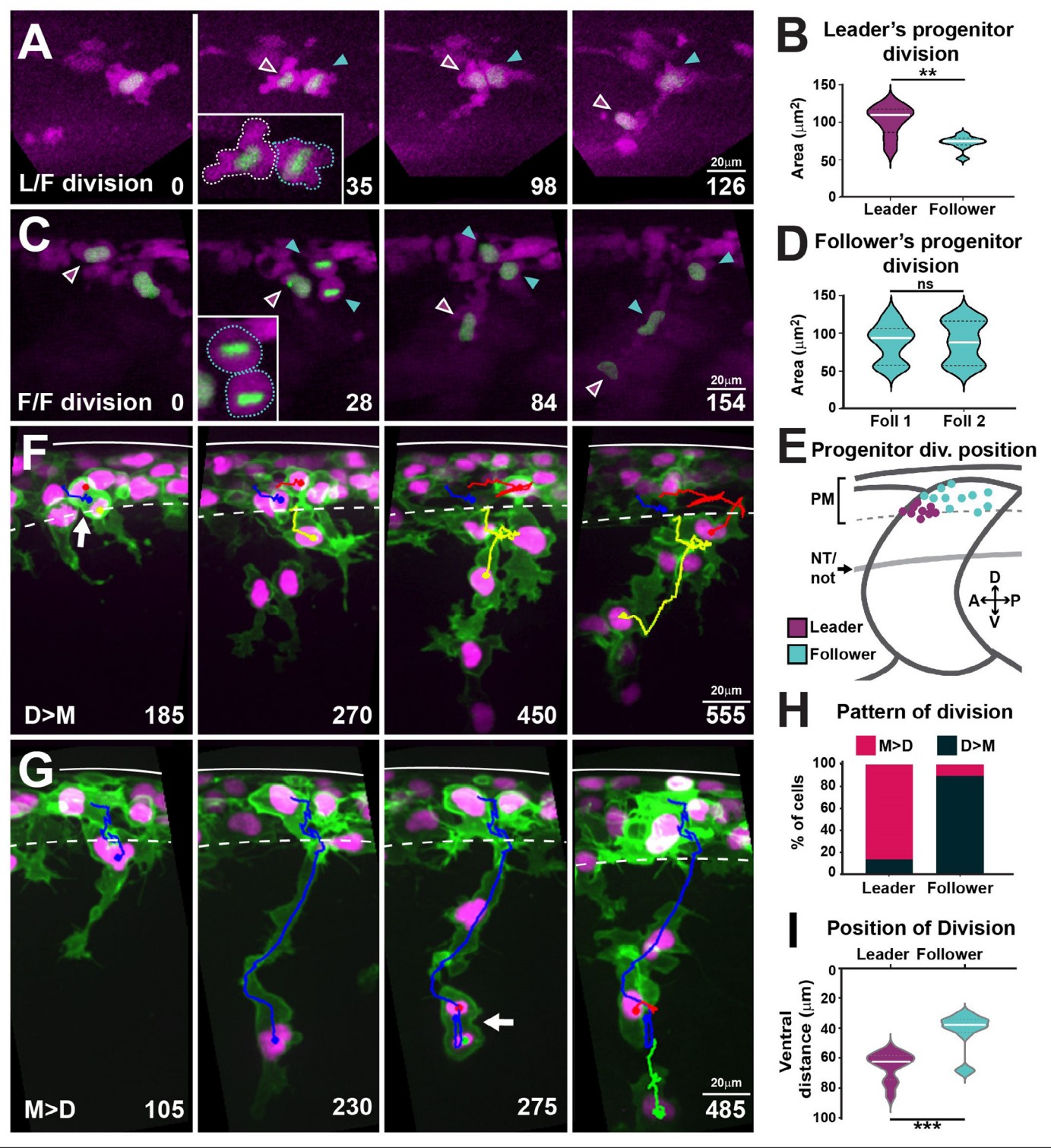

**Figure 8.** Leaders arise from the asymmetric division of a progenitor cell and present characteristic division patterns. (**A**) Selected frames from in vivo imaging of leaders' progenitor division in FoxD3:mCherry;H2aFVA:H2a-GFP embryos. (**B**) Area of leaders' progenitor daughter cells (n = 9 cells, seven embryos; Mann–Whitney *U*-test, p=0.0056). (**C**) Selected frames from in vivo imaging of followers' progenitor division in FoxD3:mCherry;H2aFVA:H2a-GFP embryos. (**D**) Area of followers' progenitor daughter cells (n = 10, four embryos; Mann–Whitney *U*-test, p>0.9999). (**E**) Position of progenitors' divisions on model embryo (leaders n = 9, seven embryos; followers n = 10, four embryos). PM, premigratory area; NT/not, neural tube/notochord boundary. (**F**) Selected frames showing the D>M division pattern from 16 to 28 hpf in vivo imaging of a Sox10:mG embryo. Blue, before division; yellow and red, after division. Arrow indicates division position. (**G**) Selected frames showing the M>D division pattern from 16 to 28 hpf in vivo imaging of a

*Figure 8 continued on next page*

*Figure 8 continued*

Sox10:mG embryo. Labelling as in (**F**). (**H**) Quantification of leaders' (n = 21, seven embryos) and follower's division patterns (n = 43, seven embryos). Red, M>D; black, D>M. (**I**) Quantification of division positions (n = 13 leaders, n = 19 followers, seven embryos; Mann–Whitney *U*-test, p=0.0002). Time in minutes. Leaders in magenta, followers in cyan. Anterior left, dorsal top.

The online version of this article includes the following video for figure 8:

**Figure 8—video 1.** Leader cells arise from the asymmetric division of a progenitor cell.

https://elifesciences.org/articles/73550/figures#fig8video1

**Figure 8—video 2.** Leader and follower cells present distinct division patterns.

https://elifesciences.org/articles/73550/figures#fig8video2

Notch signalling is a seemingly simple pathway that directly transduces receptor activation into changes in gene expression. Nevertheless, its outcomes in terms of cellular patterning are very diverse, from the generation of gene expression boundaries to temporal oscillations, or from the induction of similar fates in neighbouring cells to forcing adjacent cells into alternative fates. The latter function, known as lateral inhibition, is characterised by an intercellular negative feedback loop regulating the expression of Notch ligands. The activation of the Notch receptor in a 'signal-receiving' cell leads to the downregulation of Notch ligands expression, making it less able to act as a 'signal-sending' cell. The signature 2D patterning outcome of lateral inhibition is a mosaic of signal-sending cells with low Notch activity, surrounded by signal-receiving cells with high Notch levels. This is the case during the selection of sensory organ precursor cells in the epidermis of *Drosophila* (*Lewis, 1998*) or the formation of the mosaic of hair cells and supporting cells in the sensory organs of the inner ear (*Daudet and Żak, 2020*). In general, however, lateral inhibition operates among cells subjected to extensive rearrangements and its patterning outcome is not a salt-and-pepper mosaic of cells (*Bocci et al., 2020*). For example, during angiogenesis, cells with low Notch signalling become tip or leaders, while cells with high Notch activity differentiate as stalk or followers (*Phng and Gerhardt, 2009*). In this context, leaders are interspersed with various numbers of followers. Several models have been proposed to explain how signal-sending (leader/tip) cells can exert a long-lasting or long-range inhibition on signal-receiving (follower/stalk) cells. These take into account the modulation of Notch signalling that arise from heterogeneity in Notch receptor levels, tension, Notch-regulators, and interaction with other pathways (*Bentley and Chakravartula, 2017*; *Hadjivasiliou et al., 2019*; *Koon et al., 2018*; *Kur et al., 2016*; *Venkatraman et al., 2016*). Our data show that TNC deviate from the classical mosaic pattern, forming chains with one leader every two or three followers. Further studies will be required to define whether the aforementioned mechanisms are responsible for this architecture.

In the case of the TNC, however, the most striking divergence from the classic lateral inhibition model (or indeed angiogenesis) is the fact that the leader cell identity is associated with higher intrinsic Notch activity. In other words, there are more signal-sending cells than signal-receiving cells. This apparent inversion in the ratio of the cell types produced is surprising. Explanation of this conundrum may arise from the fact that Notch lateral inhibition, dynamics and outcomes, can be modulated by '*cis*-inhibition', a process whereby Notch ligands cell-autonomously interfere with the activation of Notch receptors (*Bray, 2016*; *del Álamo et al., 2011*). Computational models show that an increase in the strength of *cis*-inhibition can result in the inversion of the salt-and-pepper pattern (signal-sending to signal-receiving cells ratio), with the production of one cell with high Notch activity for every three cells with low Notch levels (*Formosa-Jordan and Ibañes, 2014*), a scenario that is congruent with the leader/follower ratio we observe in TNC. The detailed dynamics of lateral inhibition and whether *cis*-inhibition is at work in TNC remain to be investigated and will require direct visualisation at the single-cell level of Notch activity in live embryos.

Our data show that active progression through the cell cycle is required for TNC migration. This is consistent with studies in chicken embryos, showing that progression through $G_1/S$ is required for TNC delamination, and that NC continue cycling as they migrate (*Burstyn-Cohen*

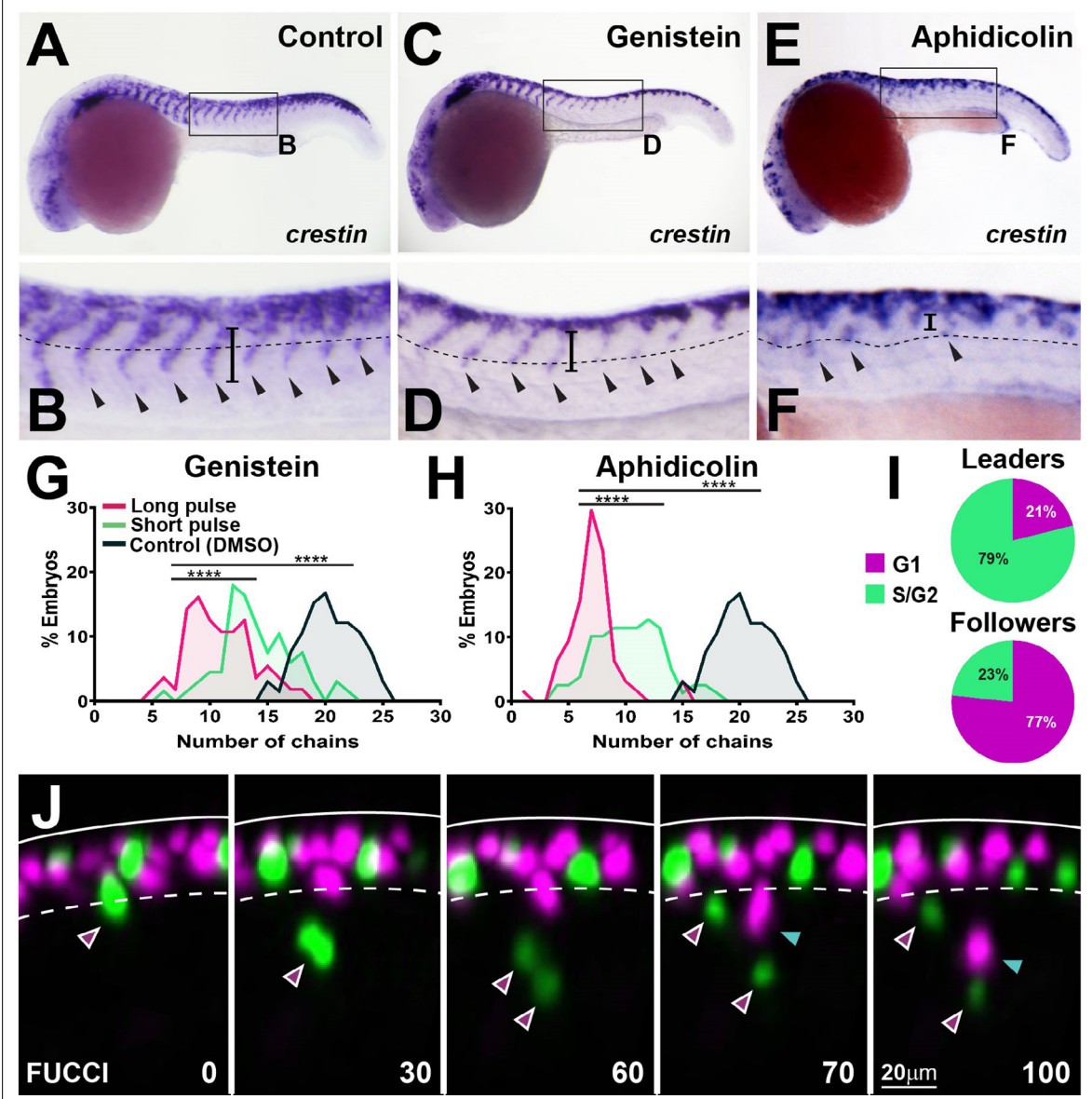

**Figure 9.** Cell cycle progression is required for trunk neural crest (TNC) migration. (**A, C, E**) *crestin* in situ hybridisation upon (**A**) DMSO, (**C**) genistein, or (**E**) aphidicolin treatment from 12 to 24 hpf. (**B, D, F**) Enlargement of areas indicated by boxes in (**A, C, E**). Dotted line marks NT/not boundary, arrowheads migratory chains, and vertical line the chain length. (**G, H**) Frequency distribution of migratory chains upon control (DMSO; n = 66), (**G**) genistein (12 hr pulse, n = 56; 6 hr pulse, n = 67), or (**H**) aphidicolin (12 hr, n = 64; 3 hr, n = 79). (**I**) Cell cycle phase at migration initiation for leaders (n = 38, four embryos) and followers (n = 43, four embryos). (**J**) Selected framed from in vivo imaging of Sox10:FUCCI. Time in minutes. Solid line marks dorsal midline, dotted line marks the premigratory area. Magenta arrowheads indicate leader and its daughters. Cyan arrowheads indicate followers.

The online version of this article includes the following video and figure supplement(s) for figure 9:

**Figure supplement 1.** Cell cycle inhibitor drugs working conditions.

**Figure 9—video 1.** Cell cycle progression is required for trunk neural crest (TNC) migration.
https://elifesciences.org/articles/73550/figures#fig9video1

**Figure 9—video 2.** Leader and follower cells initiate migration at different phases of the cell cycle.
https://elifesciences.org/articles/73550/figures#fig9video2

*and Kalcheim, 2002*; *Théveneau et al., 2007*). Our data extend these findings by showing that leader and follower cells progress through the cell cycle at different rates. Leader cells, which are larger and more motile, initiate migration in S-phase and spend twice as long in this phase as followers. It is possible that these differences arise from the fact that leaders are larger than

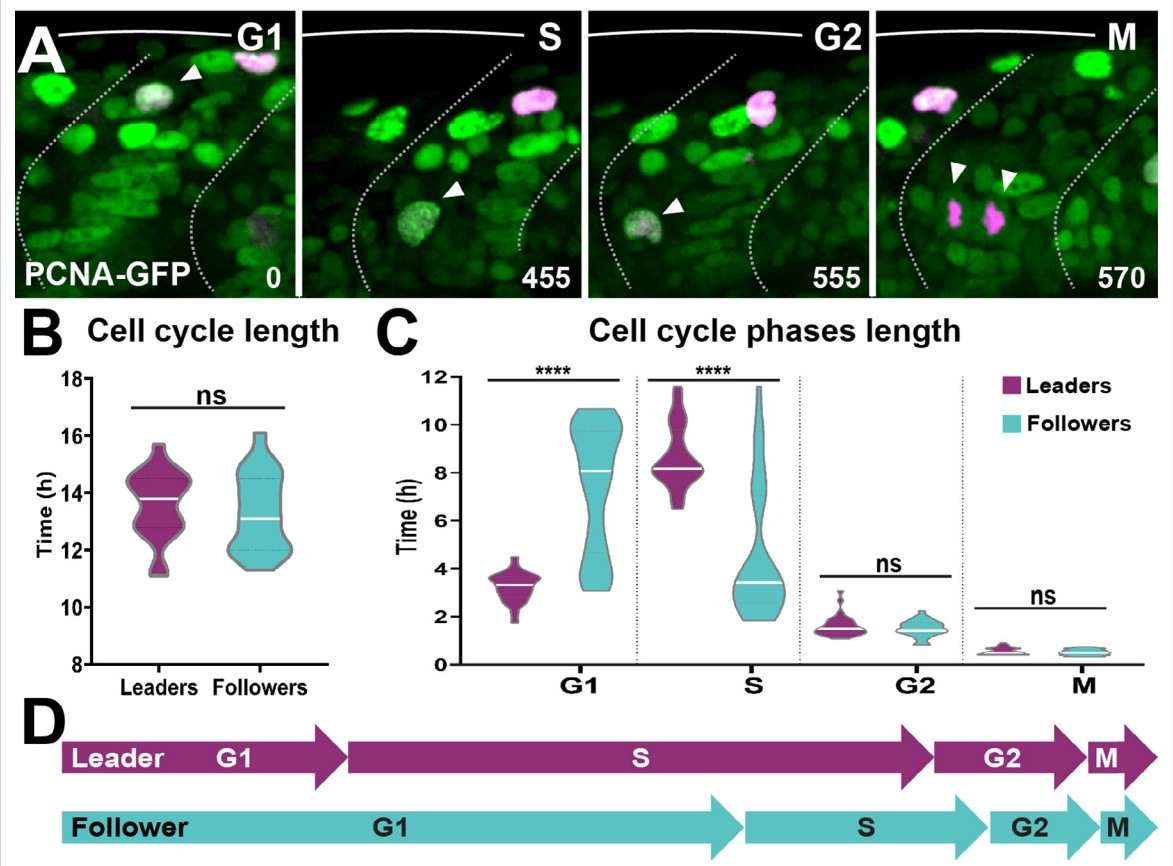

**Figure 10.** Leader and follower cells progress through the cell cycle at different rates. (**A**) Selected frames from in vivo imaging of Sox10:Kalt4 embryos from 16 to 28 hpf injected with PCNA-GFP mRNA. White arrow points to cycling cell. Time in minutes. (**B**) Quantification of the cell cycle total duration in leaders (n = 20, seven embryos) and followers (n = 19, seven embryos; unpaired *t*-test, p=0.5240). (**C**) Quantification of the cell cycle phases duration in leaders (G1 n = 45, S n = 44, G2 n = 33 and M n = 32, 11 embryos) and followers (G1 n = 50, S n = 48, G2 n = 33 and M n = 34, 11 embryos). Brown–Forsythe and Welch's ANOVA tests, G1 p<0.0001, S p<0.0001, G2 p=0.9997, M p=0.9231. (**D**) Schematic representation of the cell cycle phases durations.

The online version of this article includes the following video and figure supplement(s) for figure 10:

**Figure supplement 1.** Leader and follower cells initiate migration at distinct cell cycle phases.

**Figure 10—video 1.** PCNA-GFP reveals the cell cycle dynamics in trunk neural crest (TNC).

https://elifesciences.org/articles/73550/figures#fig10video1

followers. It has been shown that the timing of $G_1$/S transition depends on cell size and the dilution of the nuclear retinoblastoma protein (*Zatulovskiy and Skotheim, 2020*). Due to the larger volume of their cytoplasm, leader cells could be primed for a rapid G1/S phase transition. The initiation of S-phase may in turn enhance leaders' migratory characteristics through the interaction of cyclins and cyclin/CDK inhibitors (CDKI) with small GTPases. Cyclins B and D have been shown to phosphorylate cytoskeleton regulators, resulting in increased cell migration and tumour invasion (*Blethrow et al., 2008*; *Chen et al., 2020*; *Chi et al., 2008*; *Hirota et al., 2000*; *Li et al., 2006*; *Manes et al., 2003*; *Song et al., 2008*; *Zhong et al., 2010*). Furthermore, Rac1 activity, which is required for migration, oscillates during the cell cycle being highest at S-phase when cells are most invasive (*Kagawa et al., 2013*; *Walmod et al., 2004*). CDKIs, on the other hand, interact with RhoA- and ROCK-enhancing motility (*Bendris et al., 2015*; *Creff and Besson, 2020*; *Yoon et al., 2012*). Interestingly, enhanced motility increases actin branching, which in turn can accelerate the $G_1$/S transition (*Molinie et al., 2019*). These factors could therefore generate a positive feedback loop in which slightly larger leader cells

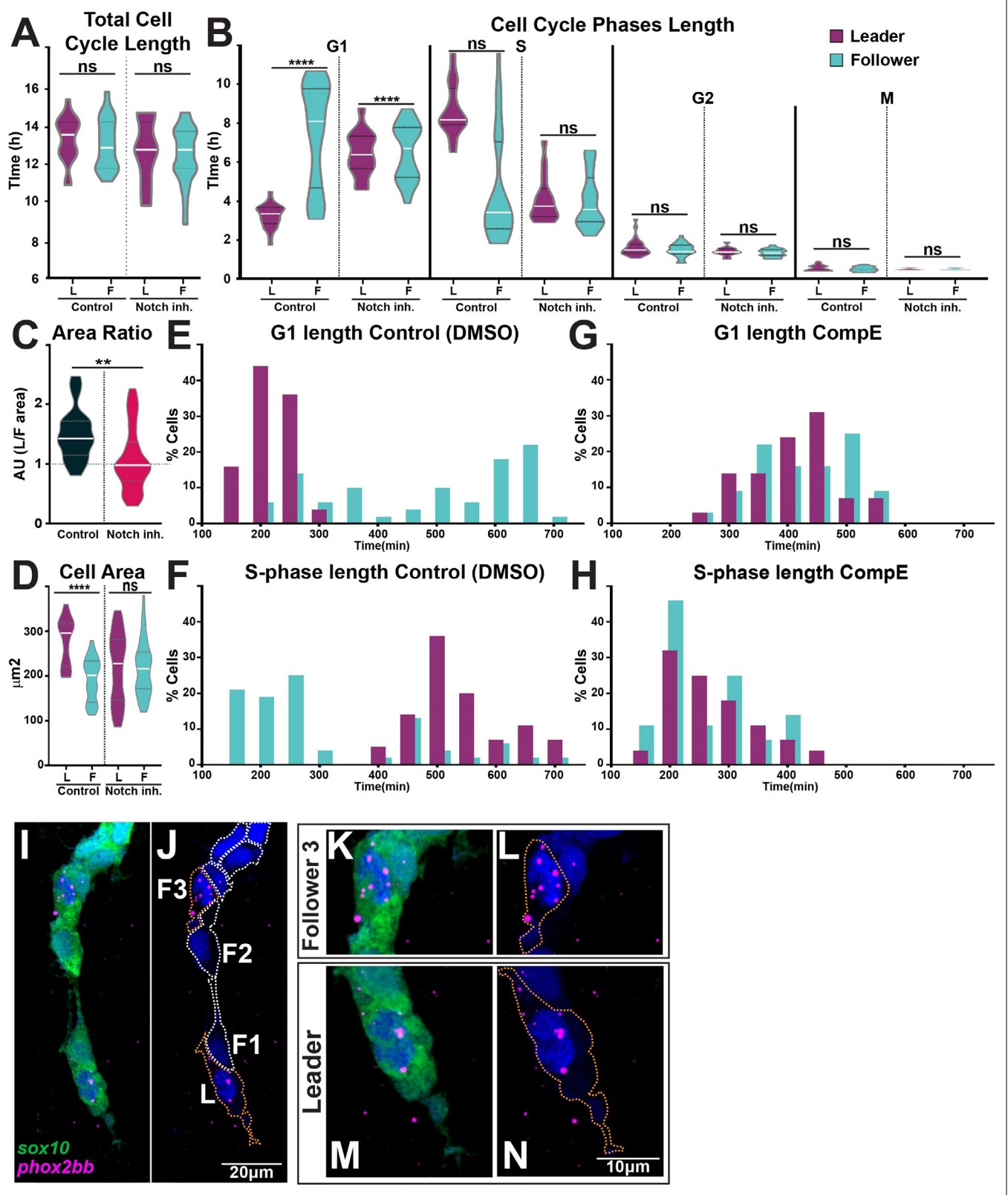

**Figure 11.** Notch signalling regulates trunk neural crest (TNC) cell cycle progression. (**A**) Quantification of the cell cycle total duration under control (DMSO, numbers as in *Figure 10B*) and Notch inhibition conditions (CompE, leaders n = 17, followers n = 22, eight embryos; one-way ANOVA, p=0.1939). (**B**) Quantification of the cell cycle phases duration under DMSO (numbers as in *Figure 10C*) and Notch inhibition conditions CompE, leaders G1 n = 29, S n = 28, G2 n = 25, and M n = 25, seven embryos; followers G1 n = 32, S n = 32, G2 n = 30, and M n = 30, seven embryos; Brown–

*Figure 11 continued on next page*

*Figure 11 continued*

Forsythe and Welch's ANOVA tests, all phases G1, S, G2, and M p>0.9999 between leaders and followers. (**C**) Quantification of cell area ratio (leaders/followers) under DMSO and Notch-inhibited conditions (n as in **D**; Brown–Forsythe and Welch's ANOVA tests, DMSO control vs. CompE p= 0.0157). (**D**) Quantification of cell area under DMSO (leaders n = 26, followers n = 22, six embryos) and CompE conditions (leaders n = 44, followers n = 41, seven embryos). Brown–Forsythe and Welch's ANOVA tests, DMSO leaders vs. followers p<0.0001, CompE leaders vs. followers p>0.9999. (**E, F**) Frequency distribution of G1- and S-phases durations in control conditions (DMSO; leaders: G1 n = 45, S n = 44, 11 embryos; followers: G1 n = 50, S n = 48, 11 embryos). (**G, H**) Frequency distribution of G1- and S-phases durations in Notch inhibition conditions (CompE; leaders: G1 n = 29, S n = 28, seven embryos; followers: G1 n = 32, S n = 32, seven embryos). (**I–N**) Images of *phox2bb* expression in 24 hpf Sox10:GFP embryo. (**K–N**) Enlargements of follower 3 and leader cells in (**I, J**). Orange dotted lines mark leader and third follower cell outline; white dotted lines mark followers' outline.

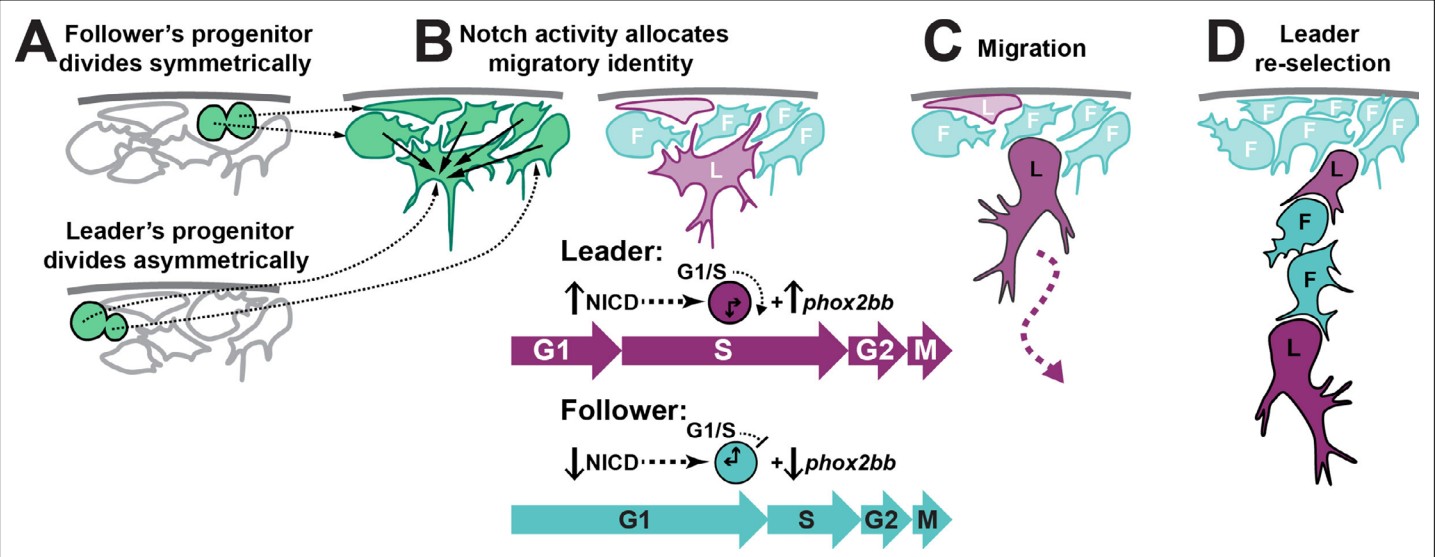

**Figure 12.** Working model of trunk neural crest (TNC) migratory identity allocation through Notch-cell cycle interaction. (**A**) Leader TNC progenitors divide asymmetrically giving rise to a prospective leader cell that is larger than the prospective followers that arise from symmetric divisions. (**B**) Interactions between TNC through Notch lateral inhibition establish higher levels of Notch activity in the bigger cell, triggering the initiation of S-phase and increased levels of *phox2bb* expression. (**C**) Leader cell initiated the chain movement while in S-phase trailed by followers in $G_1$. (**D**) Loss of the leader contact with premigratory TNC allows for a new round of Notch interaction that establishes a second leader cell.

are prone to undergo the $G_1$/S transition, in turn the activation of S-phase cyclins and CDKIs may enhance motility, reinforcing S-phase initiation.

Our data also show that TNC cell cycle progression is under the control of Notch signalling. Upon Notch inhibition, all TNC present cell cycle phase lengths typical of follower cells. Notch has been shown to regulate cell cycle in a context-dependent manner. Depending on the cell type, Notch can regulate cell cycle through the transcriptional induction of cyclins A and D, and the inhibition of CDKIs (*Campa et al., 2008*; *Dabral et al., 2016*; *Ridgway et al., 2006*; *Rizzo et al., 2008*; *Rowan et al., 2008*). Conversely, cell cycle progression can impact on Notch signalling. Notch activity is enhanced at the $G_1$/S transition, while cells become refractory to Notch during $G_2$/M (*Ambros, 1999*; *Carrieri et al., 2019*; *Hunter et al., 2016*; *Nusser-Stein et al., 2012*). Hence, the combination of large volumes and higher Notch activity levels could act synergistically to promote leaders' $G_1$/S transition.

In this study, we have uncovered new functional interactions between Notch signalling, cell cycle dynamics, and the migratory behaviour of leader and follower cells in the TNC. These complex and intricate interactions, which remain to be fully characterised at a molecular level, could apply to other cell types exhibiting collective migration. For example, studies in cancer cell lines have shown that activation or inhibition of Notch signalling hinders migration, similar to what we observe in TNC (*Konen et al., 2017*), while the maintenance of collective migration depends on the regulation of cell proliferation during angiogenesis (*Costa et al., 2016*). In view of our work, it is important to revisit the assumption that migratory phenotypes are

in conflict with cell cycle progression (*Kohrman and Matus, 2017*) and consider the possible implication for cancer therapies.

# Materials and methods

## Key resources table

| Reagent type (species) or resource | Designation | Source or reference | Identifiers | Additional information |
|---|---|---|---|---|
| Genetic reagent (*Danio rerio*) | Sox10:mG; Tg(–4.9sox10: Hsa.HIST1H2BJ-mCherry-2A-GLYPI-EGFP) | *Richardson et al., 2016* | ZDB-TGCONSTRCT-171205-3 | |
| Genetic reagent (*D. rerio*) | Sox10:Fucci; Tg(–4.9sox10:mAGFP-gmnn-2A-mCherry-cdt1) | *Rajan and Gallik, 2018* | ZDB-TGCONSTRCT-190118-1 | |
| Genetic reagent (*D. rerio*) | hs:dnSu(H); vu21Tg (hsp70l:XdnSu(H)-myc) | *Latimer et al., 2005* | ZDB-ALT-050519-2 | |
| Genetic reagent (*D. rerio*) | hs:Gal4; kca4Tg Tg(hsp70l:Gal4)1.5kca4 (1) | *Scheer and Campos-Ortega, 1999* | ZDB-ALT-020918-6 | |
| Genetic reagent (*D. rerio*) | UAS:NICD; Tg(UAS:myc-Notch1a-intra) kca3Tg | *Scheer and Campos-Ortega, 1999* | ZDB-ALT-020918-8 | |
| Genetic reagent (*D. rerio*) | Tg(UAS:dnSu(H)) | This paper | | Transgenic line details are in materials and methods |
| Genetic reagent (*D. rerio*) | Sox10:Kalt4; Tg(–4.9sox10: Hsa.HIST1H2BJ-mCherry-2A-Kalt4ER) | *Alhashem et al., 2021* | | |
| Genetic reagent (*D. rerio*) | Tg(h2afva:GFP)kca13 | *Pauls et al., 2001* | ZDB-ALT-071217-3 | |
| Genetic reagent (*D. rerio*) | Gt(FoxD3:mCherry)ct110aR | *Hochgreb-Hägele and Bronner, 2013*; *Lukoseviciute et al., 2018* | ZDB-FISH-150901-9571 | |
| Antibody | Anti-myosin heavy chain (mouse monoclonal) | Developmental Studies Hybridoma Bank | F59 | IF (1:200) |
| Antibody | Anti-synaptotagmin 2 (mouse monoclonal) | Developmental Studies Hybridoma Bank | Znp1 | IF (1:50) |
| Antibody | Anti-acetylated tubulin (mouse monoclonal) | Sigma-Aldrich | Clone 6-11B-1; Cat# MABT868 | IF (1:1000) |
| Antibody | Anti-digoxigenin-AP (sheep polyclonal) | Sigma-Aldrich | Cat# 11093274910 | IF (1:2000) |
| Antibody | Anti-GFP (chicken polyclonal) | Merck Millipore | Cat# 06-896 | IF (1:750) |
| Antibody | Anti-RFP (rabbit polyclonal) | MBL | Cat# PM005 | IF (1:750) |
| Antibody | Myc-Tag (mouse monoclonal) | Cell Signaling | Clone 9B11; Cat# 2276S | IF (1:1000) |
| Antibody | Anti-GFP (chicken polyclonal) | Thermo Fisher | Cat# A10262 | IF (1:750) |
| Recombinant DNA reagent | PCNA-GFP | Addgene | Cat# 105942 | *Leung et al., 2011* |
| Sequence-based reagent | UAS:NICD F UAS:NICD R | This paper | Genotyping primer | CATCGCGTCTCAGCCTCAC CGGAATCGTTTATTGGTGTCG 500 bp band |
| Sequence-based reagent | UAS:dnSu(H) F UAS:dnSu(H) R | This paper | Genotyping primer | GCGGTGTGTGTACTTCAGTC TCTCCCCAAACTTCCCTGTC 409 bp band |
| Sequence-based reagent | hs:dnSu(H) F hs:dnSu(H) R | This paper | Genotyping primer | CGGGCATTTACTTTATGTTGC TGCATTTCTTGCTCACTGTTTC 1 kb band |
| Commercial assay or kit | RNAscope Multiplex Fluorescent kit | Bio-Techne | Cat# 320850 | |
| Commercial assay or kit | mMESSAGE mMACHINE SP6 Transcription Kit | Thermo Fisher | Cat# AM1340 | |

*Continued on next page*

*Continued*

| Reagent type (species) or resource | Designation | Source or reference | Identifiers | Additional information |
|---|---|---|---|---|
| Chemical compound, drug | In-Fusion HD Cloning Plus | Takara | Cat# 638910 | |
| Chemical compound, drug | ProLong Gold Antifade Mountant | Thermo Fisher | Cat# P10144 | |
| Chemical compound, drug | Hydroxyurea | Sigma-Aldrich | Cat# H8627 | 20 µM |
| Chemical compound, drug | Aphidicolin | Sigma-Aldrich | Cat# A0781 | 300 µM |
| Chemical compound, drug | Genistein | Calbiochem | Cat# 345834 | 100 µM |
| Chemical compound, drug | Teniposide | Sigma-Aldrich | Cat# SML0609 | No effect on cell cycle in zebrafish |
| Chemical compound, drug | DAPT | Sigma-Aldrich | Cat# D5942-25MG | 100 µM |
| Chemical compound, drug | Compound E | Abcam | Cat# ab142164 | 50 µM |
| Software, algorithm | Tamoxifen | Sigma-Aldrich | Cat# H7904 | 2.5 µM |
| Software, algorithm | GraphPad Prism 9 | GraphPad Software | | |
| Software, algorithm | Fiji | ImageJ | *Schindelin et al., 2012* | |

## Resource availability

Further information and requests for resources and reagents should be directed to and will be fulfilled by the lead contact, Claudia Linker (claudia.linker@kcl.ac.uk).

## Materials availability

Newly generated materials from this study are available by request from the lead contact, Claudia Linker (claudia.linker@kcl.ac.uk), except for computational tools to be requested from Katie Bentley ( katie.bentley@crick.ac.uk).

## Data and code availability

The model code is accessible at https://github.com/Bentley-Cellular-Adaptive-Behaviour-Lab/Neural-CrestCpp, (copy archived at swh:1:rev:cdf63f3786390fb7905092717456cc69f5657ddc; *Alhashem, 2022b*). The code used to perform the LDA is accessible in the supplementary files. All numerical data used in the figures are accessible in the supplementary data source file.

## Zebrafish lines and injections

Zebrafish were maintained in accordance with UK Home Office regulations UK Animals (Scientific Procedures) Act 1986, amended in 2013 under project licence P70880F4C. Embryos were obtained from the following strains: *wild type, AB strain; Sox10:mG, Tg(–4.9sox10: Hsa.HIST1H2BJ-mCherry-2A-GLYPI-EGFP); Sox10:Fucci, Tg(–4.9sox10:mAGFP-gmnn-2A-mCherry-cdt1); hs:dnSu(H), vu21Tg (hsp70l:XdnSu(H)-myc); hs:Gal4, kca4Tg Tg(hsp70l:Gal4)1.5kca4 (1); UAS:NICD, Tg(UAS:myc-Notch1a-intra)kca3; Sox10:Kalt4, Tg(–4.9sox10: Hsa.HIST1H2BJ-mCherry-2A-Kalt4ER); UAS:dn-Su(H), Tg(UAS:dnSu(H)-myc); Tg(h2afva:GFP)kca13; 12XNRE:egfp.* Embryos were selected based on anatomical/developmental good health and the expression of fluorescent reporters when appropriate, split randomly between experimental groups and maintained at 28.5°C in E3 medium. Genotyping was performed by PCR of single embryos after imaging when required (UAS:NICD; UAS:dnSu(H); hs:dnSu(H)). Injections were carried at 1–4-cell stage with 30 pg of PCNA-GFP mRNA in a volume of 1 nl. mRNA was synthesised from pCS2 + PCNA GFP plasmid, kindly provided by C. Norden (IGC, Portugal), linearised with NotI and transcribed with the SP6 mMessage Machine Kit (Thermo Fisher Scientific, Cat# AM1340).

## Live imaging and tracking

Imaging and analysis were carried out as in *Alhashem et al., 2021*. In short, embryos were mounted in 1% agarose/E3 medium plus 40 µM Tricaine. Segments 6–12 were imaged in lateral views every 5′ from 16 hpf for 16–18 hr in an upright PerkinElmer Ultraview Vox system using a ×40 water immersion objective. 70 µm z-stacks with 2 µm z-steps were obtained. Image stacks were corrected using Correct 3D Drift Fiji and single-cell tracking performed with View5D Fiji plugin. Tracks were displayed using the MTrackJ and Manual Tracking Fiji plugins. Cell area measurements were done in Fiji using the free-hand selection tool to draw around cell membranes in 3D stacks using the orientation that best recapitulated the cell morphology (as in *Richardson et al., 2016*). Cell speed measurements were calculated from 3D tracks using the following formula: $((SQRT((X1-X2)^2+(Y1-Y2)^2+(Z1-Z2)^2))/T)*60$, where X, Y, and Z are the physical coordinates and T is the time step between time-lapse frames. Ventral distances were measured in a straight line from dorsal edge of the embryo to the cell position at the end of the movie. Cell directionality measurements were calculated using a previously published Excel macro (*Gorelik and Gautreau, 2014*). Total duration of the cell cycle was measured between two mitotic events. Cell cycle phase duration were measured using the characteristic nuclear pattern of PCNA-GFP, in movies where only TNC (expressing RFP and GFP) are shown using this custom Fiji macro:

```
macro "Segment Nuclei [s]" {
title = getTitle();
run("Split Channels");
selectWindow("C1-" + title); //select window with C1 in its name, nuclei
should be C1
getDimensions(width, height, channelCount, slices, frames);
run("Subtract Background...", "rolling = 200 sliding stack");
setAutoThreshold("Default dark");
run("Threshold...");
setThreshold(5, 255); //change as appropriate for your cells
setOption("BlackBackground", false);
run("Convert to Mask", "method = Default background = Dark");
run("Close");
run("Fill Holes", "stack");
run("Despeckle", "stack");
run("Dilate", "stack");
run("Dilate", "stack");
//now go over every frame and slice
for(frame = 1; frame ≤ frames; frame++){
for(slice = 1; slice ≤ slices; slice++){
 selectWindow("C1-" + title);
 setSlice(slice);
 Stack.setFrame(frame);
 run("Create Selection");
selectWindow("C2-" + title);
 setSlice(slice);
 Stack.setFrame(frame);
 run("Restore Selection");
   setBackgroundColor(0, 0, 0);
   run("Clear Outside", "slice");
```

## In situ hybridisation, immunostaining, and sectioning

The whole-mount in situ hybridisation protocol was adapted from https://wiki.zfin.org/display/prot/Whole-Mount+In+Situ+Hybridization. In short, embryos were fixed overnight (O/N) in 4% para-formaldehyde (PFA), dehydrated in 100% methanol, then rehydrated, digested with proteinase K for different times depending on the stage and pre-hybridised for 2 hr at 65°C. Riboprobes were

added, and embryos incubated at 65°C O/N. Probes were removed and embryos washed and equilibrated to PBS. Embryos were incubated in blocking solution for 2 hr and in anti-dig antibody O/N (Sigma-Aldrich, Cat# 11093274910), washed 5 × 30′ and NBT/BCIP colour reaction performed. Riboprobes for *notch1a, dlb (deltaB), dld (deltaD), her4, cb1045* were kindly provided by J. Lewis (CRUK); *crestin, mbp, bdh, myoD* by S. Wilson (UCL, UK). After the in situ colour development, embryos were processed for sections, washed 5 × 10′ with PBS, embedded in OCT, frozen by dipping the blocks in dry ice-cold 70% ETOH, and sectioned to 12–15 µm using a cryostat. Sections were thawed at room temperature (RT), incubated with blocking solution for 30′ (10% goat serum, 2% BSA, 0.5% Triton, 10 mM sodium azide in PBS) and in anti-GFP antibody ON at 4°C (Merck Millipore, Cat# 06-896). Sections were washed with PBST 5 × 5′ (0.5% Triton-PBS) and incubated with secondary antibody for 2 hr at RT, mounted in ProLong Gold Antifade Mountant (Molecular Probes, Cat# P10144) and imaged. Whole-mount antibody staining was performed in embryos fixed for 2 hr in 4% PFA, washed 4 × 10′, incubated in blocking solution for 2 hr and in primary antibodies O/N at 4°C (anti-myc, Cell Signaling, Cat# 2276S; F59 and Znp1, Developmental Studies Hybridoma Bank; acetylated tubulin, Sigma-Aldrich, Cat# MABT868). Embryos were washed 5 × 30′, incubated in secondary antibodies O/N at 4°C, washed 6 × 30′, and mounted in 1% agarose for imaging. Imaging of sectioned and whole-mount antibody-stained samples was performed in PerkinElmer Ultraview Vox system.

RNAScope (RNAscope Fluorescent Multiplex Reagent Kit, Cat# 320850) experiments were performed as in *Alhashem et al., 2022a*. In short, embryos were fixed with 4% PFA O/N at 4°C and dehydrated in 100% methanol and stored at –20°C until processing. All methanol was removed, and embryos were air dried at RT for 30′, permeabilised with Proteinase Plus for 10′ at RT (provided in kit), washed with PBS-Tween 0.01%, and incubated with probes for *egfp* and *sox10* or *phox2bb* at 1:100 dilution at 60°C O/N. Probes were recovered, embryos washed three times with SSCT 0.2× for 15′. We followed manufacturer's instructions for amplification steps AMP 1–3 and HRP C1–C4. Opal dyes 520, 570, and 650 (Akoya Biosciences, Cat# FP1487001KT, Cat# FP1488001KT, and Cat# FP1496001KT) were added at 1:3000 dilution followed by HRP blocker. Washes in between steps were performed with SSCT 0.2× for 10′ twice. Primary a-GFP-chicken (1:750) and a-RFP-rabbit (1:750; TFS, Cat# A10262, and MBL, Cat# PM005) antibodies diluted in blocking solution (PBS-Tween 0.1%, goat serum 5%, DMSO 1%) were added and incubated O/N at 4°C. Samples were washed three times in PBS-Tween 0.1% for 1 hr and then incubated in secondary antibodies, a-chicken-Alexa Fluor488 and a-rabbit-Alexa Fluor546 (TFS, Cat# A11039 and Cat# A11010) both in a 1:1000 dilution in blocking solution, for 3 hr at RT. Samples were washed six times with PBS-Tween 0.1% for 30′. For counterstaining, DAPI was added (1:1000) in the third wash (Roche, Cat# 10236276001, 2 mg/ml). Embryos were cleared in 50% glycerol/PBS an mounted in glass-bottom Petri dishes and imaged using Zeiss Laser Scanner Confocal Microscope 880 (405, 488, 514, 561, and 633 lasers).

## Drug treatments and gene expression induction

Embryos were treated by adding cell cycle inhibitors to the media from 11 hpf and incubated for 3–12 hr at 28.5°C. 20 µM hydroxyurea (Sigma-Aldrich, Cat# H8627), 300 µM aphidicolin (Sigma-Aldrich, Cat# A0781), 100 µM genistein (Calbiochem, Cat# 345834), teniposide (Sigma-Aldrich, Cat# SML0609), or 1% DMSO as control (Sigma-Aldrich, Cat# D8418). Notch signalling was inhibited at 11 hpf by adding 100 µM DAPT (Sigma-Aldrich, Cat# D5942-25MG) or 50 µM of Compound E (Abcam, Cat# ab142164). The latter reagent was used to perform live imaging, which is difficult to do with DAPT as it generates an interfering precipitate. 1% DMSO was added as control. Gene expression was induced by addition of 2.5 µM of tamoxifen (Sigma-Aldrich, Cat#H7904) to the media at 11 hpf of Sox10:Kalt4 embryos, or by heat shock at 11 hpf in hs:Gal4 and hs:dnSu(H) embryos by changing the media to 39°C E3, followed by 1 hr incubation at this temperature, thereafter embryos were grown at 28.5°C to the desired stage.

## Generation of UAS:dnSu(H) transgenic line

Using the Infusion cloning system (Takara), the following construct was inserted into the Ac/Ds vector (*Chong-Morrison et al., 2018*): 5xUAS sequence (Tol2Kit, http://tol2kit.genetics.utah.edu/index.php/Main_Page) flanked at the 3′ and 5′ ends by rabbit β-globin intron sequence. At the 3′ end, GFP followed by SV40polyA sequence was cloned to generate the Ac/Ds dUAS:GFP vector. The *cmlc2:egfp* transgenesis marker (Tol2Kit) was cloned after GFP in the contralateral strand to prevent

interaction between the UAS and the cmnl sequences. The *Xenopus* dnSu(H)-myc sequence (*Latimer et al., 2005*) was cloned into the Ac/Ds dUAS:GFP vector at the 5' end of the 5xUAS sequence, followed by the SV40polyA sequence (*Figure 4—figure supplement 2*). Transgenesis was obtained by injecting Sox10:Kalt4 embryos with 1 nl containing 50 pg of DNA plus 30 pg of Ac transposase mRNA at 1-cell stage. Embryos carrying the transgene were selected by heart GFP expression at 24 hpf. Upon Gal4ER activation by tamoxifen, dnSu(H)-myc protein was readily detected with anti-Myc antibody (*Figure 4—figure supplement 2*). GFP fluorescence driven by UAS was never observed.

## Statistical analysis

All graphs and statistical analysis were carried out in GraphPad Prism 9. All numbers in the texts are mean ± standard deviation. Every sample was tested for normality using the d'Agostino–Pearson, followed by Shapiro–Wilk tests. Samples that passed both tests were compared using either unpaired two-tailed *t*-test or one-way ANOVA. Those without a normal distribution were compared through a Mann–Whitney *U*-test, Kruskal–Wallis test, or Brown–Forsythe and Welch's ANOVA tests. For all analyses, p-values < 0.05 were deemed statistically significant, with ****p<0.0001, ***p<0.001, **p<0.01, and *p<0.05. Full statistical analysis of data in *Figure 5* is presented in *Supplementary file 1*.

## Computational model

The computational model used in this study is described in Appendix 1.

Standard LDA was carried out using the sklearn package in Python (see supplementary code files).

## Acknowledgements

In memory of Julian Lewis. We are especially grateful to N Daudet for his scientific and personal support. To the KCL fish facility staff, particularly to J Glover. We are grateful to Y Hinits and S Wilson for sharing reagents. This project was funded by MRC G1000080/1, Royal Society 2010/R1 and Wellcome Trust 207630/Z/17/Z to CL; MR was supported by the Eunice Kennedy Shriver National Institute of Child Health & Human Development of the National Institutes of Health under awards T32HD055164 and F31HD097957. KB and DF-B were supported by the Francis Crick Institute core funding from CRUK (FC001751), MRC (FC001751), and Wellcome Trust (FC001751); BBSRC BB/S015906/1 to RNK.

## Additional information

### Funding

| Funder | Grant reference number | Author |
| --- | --- | --- |
| Medical Research Council | G1000080/1 | Claudia Linker |
| Royal Society | 2010/R1 | Claudia Linker |
| Wellcome Trust | 207630/Z/17/Z | Claudia Linker |
| Eucine Kennedy Shiver National Institute of Child Health & Human Development of the National Institutes of Health | T32HD055164 | Manuel Rocha |
| Eucine Kennedy Shiver National Institute of Child Health & Human Development of the National Institutes of Health | F31HD097957 | Manuel Rocha |
| Cancer Research UK | FC001751 | Dylan Feldner-Busztin Katie Bentley |
| Medical Research Council | FC001751 | Dylan Feldner-Busztin Katie Bentley |

| Funder | Grant reference number | Author |
| --- | --- | --- |
| Wellcome Trust | FC001751 | Dylan Feldner-Busztin<br>Katie Bentley |
| Biotechnology and Biological Sciences Research Council | BB/S015906/1 | Robert N Kelsh |

The funders had no role in study design, data collection and interpretation, or the decision to submit the work for publication. For the purpose of Open Access, the authors have applied a CC BY public copyright license to any Author Accepted Manuscript version arising from this submission.

### Author contributions

Zain Alhashem, Conceptualization, Data curation, Formal analysis, Investigation, Methodology, Validation, Visualization; Dylan Feldner-Busztin, Conceptualization, Data curation, Formal analysis, Software, Visualization; Christopher Revell, Software; Macarena Alvarez-Garcillan Portillo, Karen Camargo-Sosa, Tatianna Corbeaux, Investigation; Joanna Richardson, Manuel Rocha, Martina Milanetto, Francesco Argenton, Natascia Tiso, Resources; Anton Gauert, Software, Visualization; Robert N Kelsh, Supervision; Victoria E Prince, Supervision, Writing - original draft, Writing - review and editing; Katie Bentley, Software, Supervision; Claudia Linker, Conceptualization, Data curation, Formal analysis, Funding acquisition, Investigation, Methodology, Project administration, Resources, Supervision, Validation, Visualization, Writing - original draft, Writing - review and editing

### Author ORCIDs

Zain Alhashem http://orcid.org/0000-0002-8320-3836
Christopher Revell http://orcid.org/0000-0002-9646-2888
Joanna Richardson http://orcid.org/0000-0003-2092-3876
Anton Gauert http://orcid.org/0000-0002-3013-5374
Francesco Argenton http://orcid.org/0000-0002-0803-8236
Natascia Tiso http://orcid.org/0000-0002-5444-9853
Robert N Kelsh http://orcid.org/0000-0002-9381-0066
Claudia Linker http://orcid.org/0000-0003-2028-6109

### Ethics

Zebrafish were maintained in accordance with UK Home Office regulations UK Animals (Scientific Procedures) Act 1986, amended in 2013 under project license P70880F4C.

### Decision letter and Author response

Decision letter https://doi.org/10.7554/eLife.73550.sa1
Author response https://doi.org/10.7554/eLife.73550.sa2

## Additional files

### Supplementary files

- Supplementary file 1. Statistical analysis of migratory parameters.
- Transparent reporting form
- Source code 1. In vivo LDA code.
- Source code 2. In silico LDA code.
- Source data 1. Figures source data.

### Data availability

The model code is accessible at https://github.com/Bentley-Cellular-Adaptive-Behaviour-Lab/Neural-CrestCpp, (copy archived at swh:1:rev:cdf63f3786390fb7905092717456cc69f5657ddc). The code used to perform the LDA analysis is accessible in the supplementary files. All numerical data used in the figures is accessible in the supplementary data source file.

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

## Appendix 1

### Computer modelling methods

A minimal discrete element model of TNC migration was developed in which each cell is modelled as an infinitesimal particle moving in 2D space. A network of neighbours within the particle system is identified by a Delaunay triangulation (*Appendix 1—figure 1*).

$$V\left(r\right) = D_e\ \left(e^{-2a\left(r-r_e\right)} - 2e^{-a\left(r-r_e\right)}\right),\quad a = \ \sqrt{k/\ 2D_e} \tag{1}$$

Thus defined, the system exhibits Brownian dynamics as described by the over-damped Langevin equation (*Equation 2*), such that the velocity of each particle is proportional to the resultant force applied to it ($\nabla \vee$), plus a stochastic component ($\zeta$).

$$\dot{\underline{x}} = \ -\ \frac{\Delta V}{\gamma} + \ \underline{\zeta} \tag{2}$$

Components of the resultant force on each cell arise from cell-cell interactions, cell-boundary interactions, and cell-autonomous motion.

### Tissue environment (boundary)

Cells move into permissive space between the neural tube/notochord and the somites. Boundary locations are specified before any simulation. The boundary is implemented as a region of space that applies strong repulsion to nearby cells (*Appendix 1—figure 1*). Any cell that moves within a cell radius of the boundary experiences a force given by the gradient of the same Morse potential used in cell-cell interactions, such that the repulsion of any cell from the boundary depends upon the cell volume exclusion and increases exponentially as the cell approaches the boundary (*Appendix 1—figure 1*).

The size and shape of the boundary represent a space for the pre-migratory cells at the top, a space in the middle where the notochord and neural tube meet (midline) and a vertical space where the chain can proceed downwards. The dimensions of the environment boundary were calibrated to in vivo measurements (*Appendix 1—figure 1*, showing micron-scale dimensions on the boundary).

The system is setup in a 'T' shape, which is interrupted in the middle by a space of horizontal mobility, because in vivo cells regularly move into this space. The wider region at the top represents the premigratory zone (PMZ) at the top of each migratory chain. Cells are able to filter in from the sides to mimic the continuous clustering of cells above migration chains.

### Cell properties/behaviours

#### Contact inhibition and autonomous motion

Cells exhibit autonomous motion in a direction determined by their internal polarisation. This polarisation is influenced by interaction with the cell's neighbours, such that the cell will try to move into empty space. We introduce contact inhibition into the model as a term in the Langevin equation (*Equation 2*), with magnitude determined by a user-defined parameter. The direction of autonomous magnitude for a given cell is found by identifying all adjacent nearest neighbours surrounding the cell, calculating the angle subtended by each adjacent pair, and bisecting the largest such angle (*Appendix 1—figure 1*). The magnitude of this autonomous velocity component is proportional to the user-defined parameter (aMag) and the square of the maximum subtended angle, representing the combined effect of greater polarisation and more free space to move into. Any cell that moves beyond a threshold distance from its nearest neighbour will stop autonomous motion, modelling the loss of polarisation when losing contact with neighbouring cells.

#### Cell volume exclusion

Cells exhibit volume exclusion (two cells repel from one another if they get closer than an equilibrium distance). This simply models how two cells cannot occupy the same space at the same time. The extent to which volume exclusion is exhibited can be thought of as the level of cell stiffness. Low $k$ means cells are squishier. This is modelled using the $k$ term in the Morse potential calculation (*Equation 1*).

## Co-attraction (co-A)

When cells drift more than the equilibrium distance apart, they are drawn back towards their neighbours with a force calculated by the Morse potential curve (*Equation 1*).

## Migratory identity

Leader and follower migratory identities were allocated to cells according to the order in which these enter the chain. That is, the first cell becomes leader then the next X many cells become follower cells before the cell after that becomes leader. A sensitivity analysis on leader cells frequency was performed by spacing parameter S.

## Simulation procedure

The simulation follows the process steps listed below *Appendix 1—figure 2* and was simulated on CAMP – the Francis Crick Institute's Linux-based high-performance computing system. Parameter combination/experimental condition pairs were run 100 times in parallel across 10 nodes.

1. Initialise cells within PMZ
2. Loop until maximum time has been reached:
    2.1 Perform Delaunay triangulation on cells
    2.2 Identify neighbours
    2.3 Calculate forces between nearest neighbours
    2.4 Apply boundary forces to all cells
    2.5 Calculate autonomous motion velocties as determined by contact inhibition
    2.6 Calculate noise
    2.7 Update system
    2.8 Apply experimental conditions for the next time step

### Model pseudocode overview.

A predefined number of cells is initialised in the premigratory zone (PMZ). Thereafter, the system enters a loop for every time step up to tmax. In this loop, forces are linearly summed to obtain each cell's velocity vector for that time step: (1) a Delaunay triangulation is performed on cells. (2) Each cell's nearest neighbours are identified. (3) Local forces between cells are calculated according to the Morse potential (*Appendix 1—figure 1*). (4) Boundary forces are applied to each cell. (5) Autonomous motion and contact inhibition are calculated for each cell. (6) Gaussian noise is added to each cell's velocity vector. (7) The system's clock is updated, as well as each cell's position. (8) Experimental conditions are applied for the next time step (e.g. giving certain cells leader qualities).

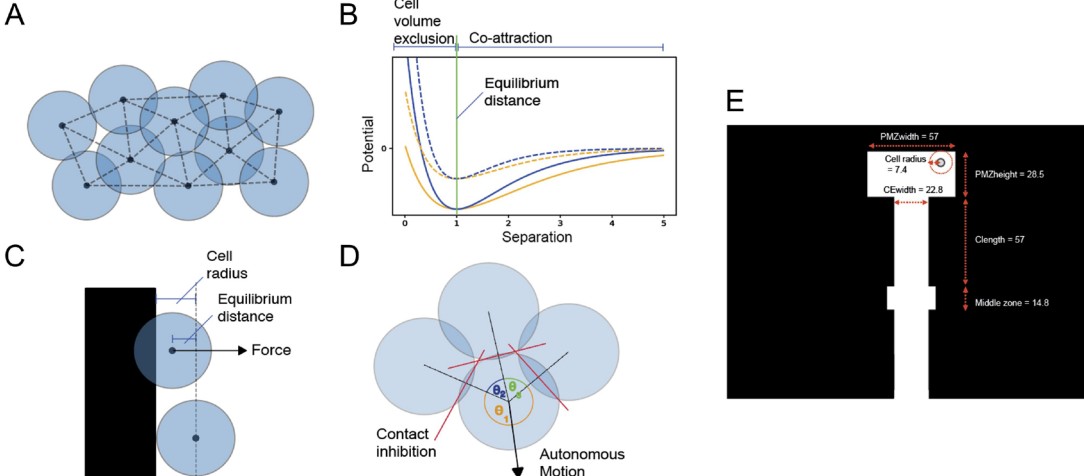

**Appendix 1—figure 1.** Description of model mechanisms and configuration. (**A**) Diagram of 10 cells modelled as infinitesimal particles, with Delaunay triangulation showing nearest neighbours and circles showing typical cell radii around each particle. (**B**) Morse potential for low-volume exclusion k (orange), high cell volume exclusion (blue), high-energy depth De (solid line), and low-energy depth (dashed line). The portion of the curve that relates to

*Appendix 1—figure 1 continued on next page*

*Appendix 1—figure 1 continued*

repulsion is distinguished from the portion that relates to attraction by the vertical green line. (**C**) Demonstrating calculation of force component from a boundary. When the centre point of a cell moves within a cell radius of the boundary, the cell experiences a force perpendicular to and away from the boundary with magnitude determined by a Morse potential and with offset from equilibrium distance. (**D**) Demonstrating calculation of cell polarisation. Adjacent nearest neighbours of a cell subtend angles $\theta 1$, $\theta 2$, and $\theta 3$ around the cell centre. The direction of polarisation, and hence autonomous motion, bisects $\theta 3$, the largest such angle. Forces on each cell arise from interactions between neighbouring particles. These interactions are defined by a Morse potential (**Morse, 1929**), a function of the separation between particles, and parameterised by an equilibrium separation (re), approximate spring constant (k), and energy depth (De) (**Equation 1**, **Figure 1B**). These parameters model the typical radius of a cell, its volume exclusion, and chemoattractive magnitude ('co-attraction'). (**E**) Dimensions of the model. White space represents empty space where cells can move freely, black space is space where cells cannot move due to boundaries. Horizontal movement is restricted while moving down the chain except for in the middle zone (for values associated with these parameters, see **Appendix 1—table 1**).

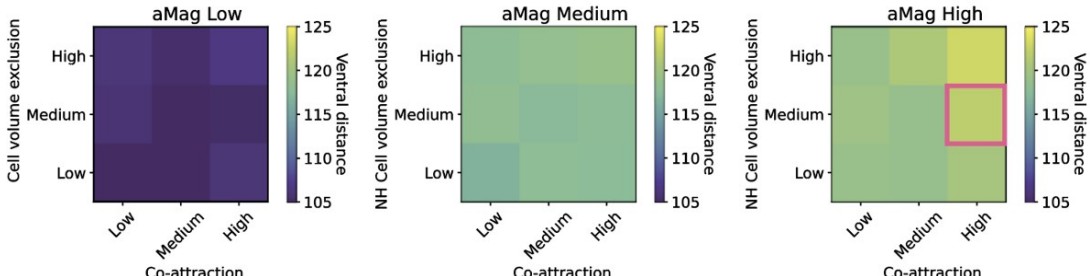

**Appendix 1—figure 2.** Calibration on final position of the furthest travelling cell in µm. The optimal distance is 120 µm which is shown by the pink square. Ventral distance increases with increases in cell volume exclusion and co-attraction, which is most apparent in the rightmost heatmap.

## Parameterisation

Time was calibrated as follows: in vivo control cells tend to migrate to approximately 120 µm from dorsal midline on average (**Figure 5C**). The total time of migration is on average 11.64 hr long (~700 min). In 2000 time steps, the control case (with differential CIL, heterogenous migratory identities and S = '1:3') also migrates to approximately 120 µm. We gathered data every 20 time steps, which means in our simulation movies there are 100 frames (i.e. 1 frame = 7 min).

Where possible, parameters were calibrated to values measured in vivo (**Appendix 1—table 1**). Model-specific parameters unable to be linked directly to in vivo values were set to values that produced realistic bounds of behaviour.

**Appendix 1—table 1.** Simulation parameters, description, range, and source.

| Name | Description | Range | Optimised setting | Units | Source |
|---|---|---|---|---|---|
| PMZ width | Horizontal space of the premigratory zone. | 57.0 | 57.0 | µm | Measurement |
| PMZ height | Vertical space in the premigratory zone. | 28.5 | 28.5 | µm | Measurement |
| CE width | Horizontal width in the migratory chain. | 22.8 | 22.8 | µm | Model specific |
| MZ ratio | Vertical space around the midpoint relative to the height of the PMZ. | 0.5 | 0.5 | Units | Model specific |
| Cell radius | Interaction radius of cell radius was inferred assuming cells were perfect spheres, based on volumetric measurements (**Richardson et al., 2016**). | 7.4 | 7.4 | mm | Measurement |
| Nc | Number of cells. | 18 | 18 | Number | Measurement |
| $\zeta$ | Magnitude of stochastic component. Term of the Langevin equation, which controls random cell movement magnitude. | 0.035 | 0.035 | Units | Model specific |
| $\gamma$ | Overdamped Langevin equation drag factor. | 1 | 1 | Units | Model specific |

*Appendix 1—table 1 Continued on next page*

*Appendix 1—table 1 Continued*

| Name | Description | Range | Optimised setting | Units | Source |
|---|---|---|---|---|---|
| S | Leader spacing – number of follower cells between leader cells in migration. | {0, 1, 2, 3, ∞} | 3 | Number | Calibrated |
| Follower k | Spring constant near equilibrium (parameter of Morse potential) for follower type cells. This can be thought of as the cell volume exclusion of the cells. High k means that cells are stiffer. | Low: [0.01] Medium: [0.02] High: [0.03] | 0.01 | Units | Calibrated |
| Leader k | As above but for leader-type cells. | Low: [0.01] Medium: [0.02] High: [0.03] | 0.02 | Units | Calibrated |
| Follower De | Depth of potential well (parameter of Morse potential). Greater De means greater range of co-attraction. This can be thought of as the amount of chemotactic attraction signal released by each cell. | Low: [3e-05] Medium: [6e-05] High: [9e-05] | 3e-05 | Units | Calibrated |
| Leader De | As above but for leader-type cells. | Low: [3e-05] Medium: [6e-05] High: [9e-05] | 6e-05 | Units | Calibrated |
| Follower aMag | Magnitude of autonomous cell velocity. In the model's implementation of contact inhibition, cells move into the widest open space. This parameter modulates the velocity with which they move into this space. | Low: [1.1e-07] Medium: [1.56e-06] High: [3e-06] | 1.1e-07 | Units | Calibrated |
| Leader aMag | As above but for leader-type cells. | Low: [1.1e-07] Medium: [1.56e-06] High: [3e-06] | 3e-06 | Units | Calibrated |
| Interaction threshold | Multiples of cell radii beyond which neighbours no longer cause polarisation by contact inhibition. | 1 | 1 | Units | Model specific |
| T max | Total run time in arbitrary units. | 2000 | 2000 | Units | Model specific |
| dt | Time interval between iterations. | 0.1 | 0.1 | Units | Model specific |
| Output interval | Time interval between data outputs. | 10 | 10 | Units | Model specific |

## Sensitivity analysis

In the grid search calibration approach, we fixed follower cells' properties to be at their low levels. Next, we looked at how changes to leader cell physical properties affected ventral distance (*Appendix 1—figure 2*). This shows a strong effect in leader aMag, whereby low leader aMag resulted in cells not migrating much beyond 100 µm no matter the level of co-attraction or cell volume exclusion. aMag had to be varied across a wider range to see a clear effect. Through this, aMag has a dominating effect on ventral distance: higher aMag is associated with higher ventral distance.

