## [Editor Report]

Using a combination of in vivo and in silico approaches, the authors have demonstrated how cell-fate decisions are orchestrated at the level of leader vs. follower cells in collective cell migration of trunk neural crest cells. They highlight the role of Notch signaling and cell cycle progression, showing how these traits differ between the leader and follower cells. The findings are of wide interest, as collective cell migration is a fundamental process critical for embryonic development as well as invasion of various cancers.

---

## [Decision Letter]

[Editors' note: this paper was reviewed by Review Commons.]

---

## [Author Response]

Reviewer # 1:

We are delighted that this reviewer finds that our work builds an elegant story that sets the stage and provides an effective conceptual framework for future explorations. Indeed, the reviewer points out the importance of TNC-environment interaction for migration, and we are actively working on this question in a follow-up manuscript. Beyond this, the reviewer does not raise any specific questions.

Reviewer # 2:

We are pleased that in the reviewer’s opinion we tackle important and challenging questions, she/he finds our study is well-designed and appreciates the interdisciplinary approach taken.

1. In Figure 3F-H, the authors present how one parameter at a time (cell speed, cell directionality) is different for leader vs. follower in different scenarios. This is a good starting point for such analysis but does not reveal relative contributions or causal connection of these parameters to the leader-follower decision. The authors should perform some kind of dimension reduction analysis (PCA, LDA) to identify which changes in their model can trigger a transition from leader to follower or vice versa. Can these predictions be tested experimentally in the TNC? Moreover, the authors comment on the role of heterogeneity in these parameters to be able to recapitulate experimental observations. Is this heterogeneity essential in all parameters here?

As suggested by the reviewer we performed a linear discriminant analysis (LDA). First, we analysed the in vivo data to assess the relative contribution of Speed, Directionality and Ventral distance to migratory identity. A random dataset drawn from a normal distribution was included as a negative control. Any association shown by LDA that is no greater than the random coefficient can be considered as irrelevant. Importantly, the LDA projection successfully separated the samples (Author response image 1) with a classifier accuracy of 80% (defined as number of correct predictions/total predictions; Author response image 1). The LDA coefficient shows that all three measurements (ventral distance, directionality and speed) have a positive association with cell type, in that order of importance (Author response image 1). Next, we use LDA to interpret our model simulations. As for the in vivo case, LDA successfully separated the samples (Author response image 1), but in this case with an elevated classifier accuracy of94% (Author response image 1). In the case of the model, we could not directly use LDA to determine which were the most important parameters in determining migratory identities, as leader or follower types are assigned at the beginning of the simulations. Instead, as suggested by the reviewer, we analysed the importance of heterogeneity between leaders and followers for each of the model parameters. LDA shows that CIL intensity is the parameter that differs most between leader and follower cells, while heterogeneity is not essential for other parameters is the only parameter in which heterogeneity is essential to match in vivo behaviour (Author response image 1). A summary of these data is now included in the Results section of the paper (page 7 line 31 onwards; Figure 7).

**Author response image 1. sa2fig1:** LDA results. A. LDA leader/follower class separation of in vivo data. B. LDA leader/follower class separation of in-sillico data. C. LDA matrix of class separation confusion of in vivo data. D. LDA matrix of class separation confusion of in-sillico data. E. LDA coefficients of in vivo measurements. F. LDA coefficients of in-sillico parameters..

2. *The in-silico analysis suggests that more than one leader cell is required for TNC migration. Has this been demonstrated by authors earlier or previous studies to be true in the case of TNC?*

The requirement of more than one leader cell in the TNC chain is a novel prediction of the model that we developed. Previous studies have therefore not addressed this point. Nevertheless, our model is supported by the following experimental findings:

A. In Figure 10 E-G the frequency distribution of the cell cycle phases’ lengths was analyzed. The follower population presents a bimodal distribution in which the minor mean coincides with the leader population mean, indicating the presence of leader cells within the follower population. Moreover, the percentage of cells that comprise the smaller peak (G1: 26% and S: 31%) is in accordance with the model prediction of a 1:2 or 1:3 leader/follower ratio in the chain. These results are described in the text.

B. We have found *Phox2bb* to be strongly enriched in leader cells as well as being expressed by cells in the positions of follower 2 or 3. These data are included in Figure 11 of the manuscript.

3. The authors claim that cell cycle progression is required for TNC migration, and that Notch signaling is crucial for cell cycle regulation. Which aspects of TNC migration are impacted upon perturbing cell cycle and/or Notch – is it contact inhibition of locomotion, group cohesion, cell overlap or a combination of them? This causal mechanistic connection seems lacking.

Experiments presented in the paper show that inhibition of cell cycle progression halts TNC migration. Moreover, in vivo imaging of these embryos shows that perturbation of cell cycle progression does not affect motility but impairs TNC chain formation (Figure S4; Video S6). Together these data support the conclusion that cell cycle progression is required for TNC migration. Our manuscript also discusses at length the possible connections between cell cycle progression and cell migration, and how Notch signalling may regulate these processes concomitantly (Discussion, page 12 and 13).

While we agree with the reviewers that understanding the direct mechanisms by which Notch and/or cell cycle progression regulate TNC migration is not only interesting but also important, such study will require an amount and depth of work that is beyond the scope of this already extensive study. For example, to understand which aspect of TNC migration is altered upon changing the cell cycle phases length, will require the generation of new transgenic lines in which the cell cycle phase lengths can be altered exclusively in NC cells with time control and, ideally, in a reversible manner. The generation and validation of such tools, followed by long term live imaging and analysis of migratory parameters at the single cell level, entails an amount of work that will provide a full new story and, most probably open new lines of investigation. We consider such an endeavour to be beyond the scope of the present manuscript.

4. Do the authors observe no overtaking or exchange of leader and follower positions in TNC migration, as seen in Zhang et al. PNAS 2019, in absence of compound E treatment?

Our previous work (Richardson et al., 2016) shows that TNC migration requires the establishment of leader and follower cells which differ in morphology and migratory parameters (leader cells always remain at the front of the chain, are larger, move faster and more directional than followers). Moreover, we have also shown that TNC migratory identities are established before migration initiation and are not interchangeable thereafter. In accord with our previous work, herein leader/follower exchanges are not observed in control conditions. We do not find it surprising that our results differ from the migratory behaviour reported by Zhang et al., as these studies were carried out in a different model system and under different experimental conditions. Our work was performed in vivo in zebrafish TNC, while Zhang et al. analysed breast cancer cell migration in vitro and ex vivo, with cells embedded in an artificial extracellular matrix. Hence, the different behaviours observed may be due to a combination of cell type specificity and the environment in which cells migrate. Having said that, our findings and those of Zhang et al. are not necessarily in contradiction. The fact that TNC leaders are not overtaken may be explained by differences in metabolism as demonstrated by Zhang et al. Their paper established that leader cells require high energy levels to retain the front position. As cells migrate the leader gradually depletes its available energy and is overtaken by a follower with higher energy levels. Our study showed that leader cells are larger than followers (Richardson et al., 2016), and that this difference is established at birth by the asymmetric division of a progenitor cell (Figure 8 A-D). It is conceivable that larger leader cells, which bear more mitochondria, do not deplete their energy during migration, and that followers that remain smaller do not attain the higher energy levels required to overtake the leader.

Further, in cases of follower cells overtaking the leader cell as shown for compound E treatment, do cells also change their cell cycle status? If any, how and when does the cell division take place?

Our data show that upon compound E treatment leader cells are unable to retain the front position being overtaken by followers (Figure 4 and 5, Video S1) and present the characteristic cell cycle profile of follower cells (Figure 11B). Taken together these data show that leader cells that are overtaken by followers do indeed change their cell cycle status. It is important to notice that Notch inhibition does not affect the total length of the cell cycle (Figure 11A; on average Control: 13.4h ± 1.3 hours; Notch Inhibition (CompE): 12.8h ± 1.6 hours), and in accord, the total number of TNC cells present in Notch gain and loss of function conditions is not significantly different from control conditions. We have now included these data in page 10, line 19 onwards and Figure S4.

*5. The authors discuss the role of Notch-Δ mediated lateral inhibition in ensuring a leader-follower commitment. Does the lateral induction mediated by Notch-Jagged signaling (Vilchez, Bocci et al. bioRxiv 2021) not interfere here?* Have they measured Δ and Jagged levels in leader vs. follower cells in TNC to establish this claim?

Our data show that Notch activity levels in TNCs define migratory identity. Communication through Notch lateral inhibition has been shown to define binary fate decisions in many systems. Hence, we hypothesized that migratory identity is allocated through Notch lateral inhibition, and we discuss at length the mechanisms and the modulators that may be responsible for generating the leader/follower ratio observed. Among the modulators that may play a role in this process, are the expression of more than one Notch receptor and the heterogeneity in their levels of expression among TNC (page 12, line 7 onwards).

As the reviewer mentions we have not been able to measure the levels of Notch receptor expression in TNC. We agree that such an analysis is important and for this reason we invested considerable energy and resources to obtain these data, attempting in situ hybridization for Notch components and effectors (in sections and wholemount embryos), as well as analysis of the Her4:GFP reporter. However, none of these approaches produced fully satisfactory results due to the morphology of TNC cells, which are very thin and flat, and intermingled with other tissues (neural tube, somites, and epidermis) that presents high levels of Notch signalling. Prompted by the reviewers comments we have now analysed a new Notch reporter transgenic (Moro et al., 2013), and in these embryos clear differences in the levels of Notch activity between TNC can be observed (Figure 1). These data support our conclusions and are consistent with Notch lateral inhibition being the mechanism allocating TNC migratory identity. Considering that the concept of lateral induction is not required to explain our data and the absence of quantitative data on levels of Notch receptors, it is our opinion that introducing lateral induction in the discussion would be highly speculative and will lengthen and overcomplicate the manuscript.

Reviewer # 3:

The reviewer finds our study of wide interest, carefully done and finds our conclusions supported by the data presented. Nevertheless, she/he finds two moderately weak points: (1) there is no specific marker(s) to define leaders versus followers, and (2) the authors do not show expression of Notch pathway components. In the revised version of our manuscript, we address both of these points by analysing *Phox2bb* expression, a molecular marker of TNC leaders, and by studying Notch signalling levels in TNC using a Notch reporter line.

Major commentsWhile, it does not appear that Notch signaling has been examined in the context of Trunk Neural Crest (TNC) migration in zebrafish, previous studies have examined the effects of Notch signaling in mice in regards to cardiac neural crest cell migration as well as neural crest cells that contribute to the enteric nervous system (Mead and Yutzey, 2012). Researchers found that either gain or loss of function lead to deficient neural crest cell migration in both contexts leading to a lack of neural crest cell contribution to the developing cardiac outflow tract and a neural crest ell deficiency in the ENS. Additionally, these researchers (Mead and Yutzey 2012) also examined the effects of Notch signaling on neural crest cell proliferation in mice and found a loss of Notch signaling results in decreased proliferation of NCC's in the DRG. Therefore, while this manuscript does go into greater depth than Mead and Yutzey did in 2012 specifically in regards to tracking cells in vivo the findings surrounding gross effects on migration and proliferation are entirely novel. These studies should be referenced in the introduction or discussion.

We fully agree with the reviewer that the work by Mead et al., 2012 is important and, indeed, this work was cited in the introduction of our initial submission. This section has now been expanded.

Figure S1: it is not possible to see whether any of these genes (with possible exception of her4) are really expressed in the TNC. It would benefit this study greatly if the authors can perform fluorescent in situs using one of their TNC transgenic lines to definitely show which Notch pathway members are expressed in leaders and followers. They are certainly capable of imaging the TNC at a single cell resolution (e.g., Figure 4 and Videos).

As mentioned before (see reviewer*#* 2 point 5), the analysis of Notch components expression did not produce satisfactory results due to the morphology of TNC cells and the fact that these are intermingled with other tissues that present high levels of Notch signalling. Nevertheless, prompted by the reviewers comments we have added analysis of a Notch reporter line (Moro et al. 2012) that clearly shows differences in the levels of Notch activity between TNC (Figure 1). These data support our conclusions and are consistent with Notch lateral inhibition being the mechanism allocating TNC migratory identity.

Figure 2: a better explanation to as to why the authors switched γ-secretase inhibitors would be valuable here.

We thank the reviewer for pointing this out, the following explanation has now been added to the Material and Methods section. Page 20, line 33:-‘Notch signalling was inhibited at 11hpf by adding 100μM DAPT (Σ-Aldrich, Cat#D5942-25MG) or 50μM of Compound E (Abcam Cat#ab142164). The latter reagent was used to perform live imaging, which is difficult to do with DAPT as it generates an interfering precipitate.’-

Figure 1 shows that the number of TNC chains is reduced following Notch inhibition. However, this mainly affects posterior TNC chains and the effect while significant is not great (especially in case of HS:Gal4xUAS:NICD). Is it because the inhibitor (heat-shock) was applied after the anterior TNC already initiated its migration? Does it take some time for these manipulations to take effect? The authors need a better explanation here.

As the reviewer points out the induction and migration of anterior NC chains (up to somite 6, heart and vagal populations) is not affected by Notch signalling alterations. This may be due to these populations having different migratory mechanisms, not being sensitive to Notch signalling, and/or to the timing of the experiment, migration being underway when Notch was altered. In any case, heart and vagal NC do not form part of the TNC population in zebrafish (reviewed in Hutchins et al., 2018), and have not been analysed in our study. Our analysis confirms previous reports (Cornell and Eisen, 2000) showing that anterior NC induction is independent of Notch signalling (Figure 2). This explanation has now been added to the text. Moreover, our data show that Notch inhibition delays TNC migration at all antero-posterior levels of the trunk (from somites 6-29; Figure 3K, Figure 1K in the original manuscript).

It is unknown whether zebrafish heart and/or vagal NC migrate collectively using similar mechanisms as TNC, or whether these are responsive to Notch signalling. Moreover, as the reviewer points out, there may be a time delay to attain full Notch inhibition, which may allow vagal NC to migrate unaffected. Full Notch inhibition is attained only 1h after treatment (chemical and heat shock inhibition, personal communication Guidicelli F. and Lewis J.; time delay of tamoxifen treatment shown in Figure S3F). As all experiments were performed at 12hpf, to avoid alterations to NC induction (Figure 2), full Notch inhibition only take place by 13hpf, time at which vagal NC migration is already underway (Raible et al., 1992).

Figures 2 and 3: While I agree that Notch inhibition causes the leader cell to be unable to maintain its leader cell position, I do not completely agree with the implication that the "collective is now all followers". Specifically how can we be sure of this? What are the specific requirements for a cell to be labeled a "leader cell" versus a "follower cell"? Definitions of how the researchers characterize cells as leaders and followers are necessary. Ideally, the authors would have markers for leaders and follower; in not they should provide a better definition.– This explanation would also be helpful in the Notch GOF experiments in regards to live imaging of cell migration. Specifically, explaining the characteristics a leader and follower cell prior to explaining the results would more clearly support the researchers findings.Page 6 Line 40: Again, based on the above comment, the authors should be careful making this conclusion: "TNC with high Notch levels become leaders while cells with low Notch activity migrate as followers". There is no specific data showing that leaders have high levels of expression of Notch signaling via reporter expression or in situ hybridization for downstream signaling molecules and vice versa for followers.

We addressed these issues in two different ways. First, as proposed by the reviewer, we have included definitions for TNC migratory identities in the text (page 4, line 43). Second, we use *Phox2bb* expression as a molecular marker for leader cells in Notch altered conditions. These data are shown in Figure 6 and confirm our previous conclusion that high Notch activity defines the leader identity.

Page 10 Line 31-34: The fluorescent images shown in Figure 4A do not seem to indicate a difference in size of cells as depicted by the graph. A better example is necessary. Also, cell volume rather than cell area is much better indicator of cell size!

We thank the reviewer for pointing out the fact that the picture shown in Figure 4A does not seem to indicate the difference in size quantified in Figure 4C. This is due to the fact that the prospective leader cell depicted in Figure 4A at 35min has an extension in the z plane (coming out of the page) that is not visible in the z projection shown. This is clearly observed in the 3D reconstruction of this stack that has now been added as Video S4. Nevertheless, the orientation of the picture in the figure has been maintained as it the best orientation to show the plane of division. Measurements of cell area were performed using the plane that best recapitulated cell morphology, this explanation has now been added to the Materials and methods section in page 20, lane 35.

We also agree with the reviewer that analysis of cell volume is a better measurement of size than cell area. Nevertheless, calculation of cell volume from in vivo imaging presents a major technical difficulty. Accurate estimation of volume from 3D rendering requires imaging at 1mm z step, as estimations obtained from imaging with bigger z steps introduce invalidating errors. Unfortunately, imaging at 1mm z step frequency produces photobleaching, cell death and embryo death after a few hours. For this reason, all imaging was performed at 2 mm z step, which is incompatible with volumetric 3D rendering. Hence, we performed area measurements as a proxy for volume. We believe this approach is not problematic because areas are calculated to the power of 2 and volumes to the power of 3, which means the differences observed in area are an underrepresentation of the volume differences. Thus, we are confident to conclude that the leader’s progenitor divides asymmetrically giving rise to daughter cells of different sizes.

Figure 6: While the author implicated that there are differences in cell cycle progression in terms of lengths of G1 and S phase, they did not examine how this affects migratory behavior. This would helpful in emphasizing their claims that "cell cycle progression may regulate their migratory behavior".

Our conclusion that cell cycle progression is required for TNC migration is fully supported by the results presented in Figure 9 and Video S5, and we discuss the possible mechanisms that may link cell cycle to migration. While we agree with the reviewer that exploring how the cell cycle phases length affect migration is very interesting, the depth and the amount of work that such analysis requires is beyond the scope of this manuscript (see answer to Reviewer 2 point 3).

*Page 17: Line 34-41: I think this is the model rather than fact. These claims need to be toned down to indicate potential mechanisms by which these cells are initiating migration*.

As the reviewer points out this section describes the proposed working model and is not intended to outline experimental results or conclusions. Importantly, we begin this section by stating that: -“we have addressed the mechanism that establishes leader and follower identities and can propose the following model”- (page 11, line 14). Taking in account the reviewer’s comment and to emphasize the fact that we are describing a working model, we have added the following phrase at the end of this paragraph (page 11, line 27): – “This working model of TNC migration is supported by both our in vivo data and our *in-silico* modeling and provides a useful conceptual framework for future studies to build upon.”-

Minor comments:Figure 3: Parameters is spelled incorrectly

This has been changed.

References:

Cornell, R.A., Eisen, J.S., 2000. Δ signaling mediates segregation of neural crest and spinal sensory neurons from zebrafish lateral neural plate. Development 127, 2873–2882.

Hutchins, E.J., Kunttas, E., Piacentino, M.L., Howard, A.G.A., Bronner, M.E., Uribe, R.A., 2018. Migration and diversification of the vagal neural crest. Developmental Biology, The Neural Crest: 150 years after His’ discovery 444, S98–S109. https://doi.org/10.1016/j.ydbio.2018.07.004

Moro, E., Vettori, A., Porazzi, P., Schiavone, M., Rampazzo, E., Casari, A., Ek, O., Facchinello, N., Astone, M., Zancan, I., Milanetto, M., Tiso, N., Argenton, F., 2013. Generation and application of signaling pathway reporter lines in zebrafish. Mol Genet Genomics 288, 231–242. https://doi.org/10.1007/s00438-013-0750-z

Raible, D.W., Wood, A., Hodsdon, W., Henion, P.D., Weston, J.A., Eisen, J.S., 1992. Segregation and early dispersal of neural crest cells in the embryonic zebrafish. Dev. Dyn. 195, 29–42. https://doi.org/10.1002/aja.1001950104

Richardson, J., Gauert, A., Briones Montecinos, L., Fanlo, L., Alhashem, Z.M., Assar, R., Marti, E., Kabla, A., Härtel, S., Linker, C., 2016. Leader Cells Define Directionality of Trunk, but Not Cranial, Neural Crest Cell Migration. Cell Reports 15, 2076–2088. https://doi.org/10.1016/j.celrep.2016.04.067